# Simultaneous Aerosol Mass Spectrometry and Chemical Ionisation Mass Spectrometry measurements during a biomass burning event in the UK: Insights into nitrate chemistry

Ernesto Reyes-Villegas[1], Michael Priestley[1], Yu-Chieh Ting[1], Sophie Haslett[1], Thomas Bannan[1], Michael Le breton[1*], Paul I. Williams[1,2], Asan Bacak[1], Michael J. Flynn[1], Hugh Coe[1], Carl Percival[1Δ], James D. Allan[1,2]

[1]School of Earth, Atmospheric and Environmental Sciences, The University of Manchester, Manchester, M13 9PL, UK
[2]National Centre for Atmospheric Science, The University of Manchester, Manchester, M13 9PL, UK
[*]Now at University of Gothenburg, 40530 Gothenburg, Sweden
[Δ]Now at Jet Propulsion Laboratory, 4800 Oak Grove Drive, Pasadena, CA 91109, USA

*Correspondence to*: Ernesto Reyes-Villegas (ernesto.reyesvillegas@manchester.ac.uk)

**Abstract.** Over the past decade, there has been an increasing interest in short-term events that negatively affect air quality such as bonfires and fireworks. High aerosol and gas concentrations generated from public bonfires/fireworks were measured in order to understand the night-time chemical processes and their atmospheric implications. Nitrogen chemistry was observed during the bonfire night with nitrogen containing compounds in both gas and aerosol phase and further $N_2O_5$ and $ClNO_2$ concentrations, which depleted early next morning due to photolysis of $NO_3$ radicals, ceasing production. Particulate organic oxides of nitrogen (PON) concentrations of 2.8 µg.m$^{-3}$ were estimated using the m/z 46:30 ratios from AMS measurements, according to previously published methods. ME-2 source apportionment was performed to determine organic aerosol concentrations from different sources after modifying the fragmentation table and it was possible to identify two PON factors representing primary (pPON_ME2) and secondary (sPON_ME2) contributions. A slight improvement in the agreement between the source apportionment of the AMS and a collocated AE-31 Aethalometer was observed after modifying the prescribed fragmentation in the AMS organic spectrum (the fragmentation table) to determine PON sources, which resulted in an r$^2$ = 0.894 between BBOA and $b_{abs\_470wb}$ compared to an r$^2$ = 0.861 obtained without the modification. Correlations between OA sources and measurements made using Time of Flight Chemical Ionization Mass Spectrometry with an iodide adduct ion were performed in order to determine possible gas tracers to be used in future ME-2 analyses to constrain solutions. During bonfire night, high correlations (r$^2$) were observed between BBOA and methacrylic acid (0.92), Acrylic acid (0.90), nitrous acid (0.86), propionic acid, (0.85) and Hydrogen cyanide (0.76). A series of oxygenated species, chlorine compounds showed good correlations with sPON_ME2 and the low volatility oxygenated organic aerosol (LVOOA) factor during bonfire night and an event with low pollutant concentrations. Further analysis of pPON_ME2 and sPON_ME2 was performed in order to determine whether these PON sources absorb light near the UV region using an Aethalometer. This hypothesis was tested by doing multilinear regressions between $b_{abs\_470wb}$ and BBOA, sPON_ME2 and pPON_ME2. Our results suggest that sPON_ME2 does not absorb light at 470 nm while pPON_ME2 and LVOOA absorb light at 470 nm. This may inform black carbon (BC) source apportionment studies from Aethalometer measurements, through investigation of the brown carbon contribution to $b_{abs\_470wb}$.

## 1. Introduction.

Exposure to combustion aerosols has been associated with a range of negative health effects, in particular wood smoke aerosols have been shown to present respiratory and cardiovascular health effects (Naeher et al., 2007). Bonfires and fireworks are one of the main sporadic events with high emissions of atmospheric pollutants (Vassura et al., 2014;Joshi et al., 2016), even when these high emissions only last a couple of hours, high pollutant concentrations may instigate adverse

effects on human health (Moreno et al., 2007;Godri et al., 2010) and severely reduce visibility (Vecchi et al., 2008). Ravindra et al. (2003) found that the short-term exposure of air pollutants increases the likelihood of acute health effects.

Due to these adverse effects, different studies have been performed to analyse air pollution during important festivities around the world, for instance New Year's Eve celebrations (Drewnick et al., 2006;Zhang et al., 2010), the Lantern Festival in China (Wang et al., 2007) and Diwali festival in India (Pervez et al., 2016) as well as football matches such as during the Bundesliga in Mainz Germany 2012 (Faber et al., 2013). In the UK, the bonfire night festivity takes place on November $5^{th}$ to commemorate Guy Fawkes' unsuccessful attempt to destroy the Houses of Parliament in 1605 (Ainsworth, 1850). During this celebration, bonfires usually followed by fireworks, are lit domestically and on larger scale communally in public parks. Different studies have been carried out to assess the air pollution during bonfire night in the UK; for instance targeting the particle size distribution (Colbeck and Chung, 1996), investigating $PM_{10}$ concentrations in different cities around the UK during bonfires (Clark, 1997), measuring dioxins in ambient air in Oxford (Dyke et al., 1997); polycyclic aromatic hydrocarbons were measured in Lancaster, 2000 (Farrar et al., 2004) and potentially toxic elements were measured and their association to health risks was assessed in London (Hamad et al., 2015).

Receptor modelling has been widely used to determine OA sources in urban environments. However, it has been used in just a small number of studies with sporadic events of high pollutant concentrations. For instance Vecchi et al. (2008) was the first to analyse measurements taken during firework displays using positive matrix factorisation (PMF). Tian et al. (2014) did a PMF analysis of $PM_{2.5}$ components, identifying five different sources: Crustal dust, coal combustion, secondary particles, vehicular exhaust and fireworks. In Riccione, Italy, Vassura et al. (2014) determined that levoglucosan, OC, PAHs, Al, and Pb, emitted from bonfires during St. Joseph's Eve, can be used as markers for bonfire emissions.

Particulate organic oxides of nitrogen (PON), a term we use here to encompass nitro-organics and organic nitrates, have been found to absorb light near the UV region (Mohr et al., 2013) and to present potential toxicity affecting human health (Fernandez et al., 1992;Qingguo et al., 1995). PON also act as a $NO_x$ reservoir in the night-time, releasing $NO_x$ concentrations when the sun rises with the possibility of increasing $O_3$ production (Perring et al., 2013;Mao et al., 2013). PON are important components of organic aerosols; for instance Day et al. (2010), in measurements taken during winter at an urban location, found that PON concentrations accounted for up to 10% of organic matter. Kiendler-Scharr et al. (2016) concluded that, at a continental scale, PON represent 34% to 44% of aerosol nitrate. Organic oxides of nitrogen can be categorised, according to their origin, into two types; primary and secondary, primary organic nitrates are related to combustion sources (Zhang et al., 2016) such as fossil fuel (Day et al., 2010) and biomass burning emissions (Kitanovski et al., 2012;Mohr et al., 2013). Secondary organic oxides of nitrogen are produced in the atmosphere, for example when $NO_3$ reacts with unsaturated hydrocarbons (Ng et al., 2017). Nitrophenols are produced from reactions of phenols, both during the day reacting with $OH + NO_2$, and at night reacting with $NO_3 + NO_2$ (Harrison et al., 2005;Yuan et al., 2016).

The Aethalometer (Magee Scientific, USA) has been widely used to measure light absorbing carbon, proving to be a robust instrument capable to operate in a variety of environments and currently is being used at many different locations around the world. The European Environment Agency, in a technical report published in 2013 (EEA, 2013), states that there are at least 11 European countries using Aethalometers. The UK has a black carbon (BC) network comprising 14 sites covering a wide range of monitoring sites (https://uk-air.defra.gov.uk/networks/network-info?view=ukbsn) and India started in 2016 a BC network with 16 Aethalometers (LASKAR et al., 2016). Commonly, Aethalometers have been used to separate sources of light-absorbing aerosols following Sandradewi et al. (2008). The approach separates absorption from traffic, predominately resulting from BC which absorbs light in the infrared region and from wood burning, which includes BC and absorbing organic matter that also absorbs near the ultraviolet region. The Aethalometer model is based in the differences of aerosol absorption, using the absorption Ångström exponent, at specific wavelength of light chosen to perform the model.

Absorption Ångström exponent values range from 0.8-1.1 for traffic and 0.9-3.5 for wood burning (Zotter et al., 2017). It is known that Brown carbon (BrC) is organic matter capable of absorbing light near the UV region (Bones et al., 2010; Saleh et al., 2014) and that PON is a potential contributor to BrC (Mohr et al., 2013). However, the mechanistic behaviour that links this behaviour to wood burning has not completely been resolved and there may be other sources such as secondary organic aerosols that can absorb near the UV region.

Here we present analysis performed on data collected during bonfire night celebrations in Manchester UK (29th October to 10th November 2014) using a cToF-AMS and a HR-ToF-CIMS along with other instruments to measure both aerosols and gaseous pollutants with the aim of understanding the night-time chemical processes and their atmospheric implications. Very high concentrations of pollutants occurred as a result of the meteorological conditions, which presented a good opportunity to investigate the detailed phenomenon as a case study, particularly the possibility to determine PON concentrations, their nature and interaction with Aethalometer measurements.

## 2. Methods

### 2.1 Site and instrumentation

Online measurements of aerosols and gases were taken from ambient air, between 29[th] October and 10[th] November 2014, at a rooftop location at the University of Manchester [53.467° N, 2.232° W], in order to quantify atmospheric pollution during the bonfire night event on and around 5[th] November. Figure S1 shows a map with the location of the monitoring site and nine public parks where bonfire/fireworks were displayed around greater Manchester. This is the same dataset presented by Liu et al. (2017).

A compact time of flight Aerosol Mass Spectrometer (cToF-AMS, here after AMS) was used to perform 5-minute measurements of organic aerosols (OA), sulfate ($SO_4^{2-}$), nitrate ($NO_3^-$), ammonium ($NH4^+$) and chloride ($Cl^-$) (Drewnick et al., 2005). This version of AMS provides unit mass resolution mass spectra information. A High Resolution Time of Flight Chemical Ionization Mass Spectrometer (HR-ToF-CIMS, here after CIMS) was used to measure gas phase concentrations, using iodide as a reagent (Lee et al., 2014). The methodology to calculate gas phase concentrations from CIMS measurements are been described by Priestley et al. (In preparation). An Aethalometer, model AE31 (Magee Scientific), measured light absorption at seven wavelengths (370, 450, 571, 615, 660, 880 and 950 nm) and a Multi Angle Absorption Photometer (MAAP; Thermo Model 5012) measured BC concentrations (Petzold et al., 2002). $NO_x$, CO, $O_3$ and meteorology data were downloaded from Whitworth observatory (http://www.cas.manchester.ac.uk/restools/whitworth/data/), which were measured at the same location. From 31[st] October to 10[th] November, a catalytic stripper was attached to the AMS, switching every 30 minutes between direct measurements and through the catalytic stripper. These measurements were performed as part of a different experiment (Liu et al., 2017). In the present study we used the AMS data from the direct measurements only, aerosol and gas data from other instruments were averaged to AMS sampling times.

### 2.2 Source apportionment

#### 2.2.1 Aethalometer model

The aerosol light absorption depends on the wavelength and may be used to apportion BC from traffic and wood burning from Aethalometer measurements as proposed by (Sandradewi et al., 2008). The absorption coefficients ($b_{abs}$) are related to the wavelengths at which the absorptions are measured ($\lambda$) and Ångström absorption exponents ($\alpha$) with the relationship: $b_{abs} \propto \lambda^{\alpha i}$, thus the following equations can be solved:

$$\frac{b_{abs\_470tr}}{b_{abs\_950tr}} = \left(\frac{470}{950}\right)^{-\alpha_{tr}} \tag{1}$$


$$\frac{b_{abs\_470wb}}{b_{abs\_950wb}} = \left(\frac{470}{950}\right)^{-\alpha_{wb}} \tag{2}$$

$$b_{abs}(470_{nm}) = b_{abs\_470tr} + b_{abs\_470wb} \tag{3}$$

$$b_{abs}(950_{nm}) = b_{abs\_950tr} + b_{abs\_950wb} \tag{4}$$


Here, it is possible to calculate the wood burning (wb) and traffic (tr) contributions to BC at 470 and 950 nanometres (nm) as used in previous studies (Crilley et al., 2015;Harrison et al., 2012). Wavelengths 470 and 950 nm were chosen as Zotter et al. (2017) determined that using this pair of wavelength resulted in less residuals compared when using the pair of wavelengths 470-880, and 370-880 nm. Before the Aethalometer model was applied, the absorption coefficients ($b_{abs}$) needed to be

corrected following Weingartner et al. (2003) as attenuation is affected by scattering and loading variations. The following parameters were calculated: multiple scattering constant C = 3.16 and filter loading factors (f) of 1.49 and 1.28 for the wavelengths 470 and 950 nm respectively. Refer to section S3 in supplement for detailed information.

**2.2.2 Particulate Organic Oxides of Nitrogen (PON)**

Concentrations of PON were calculated following the method proposed by Farmer et al. (2010) and the considerations used

by Kiendler-Scharr et al. (2016). This method has been previously used in studies looking at aerosols from biomass burning (Tiitta et al., 2016;Zhu et al., 2016;Florou et al., 2017). Equation 5 calculates the PON fraction ($X_{PON}$), using the signals at m/z 30 and m/z 46 to calculate m/z ratios 46:30 from AMS measurements ($R_{meas}$), from ammonium nitrate calibrations ($R_{cal}$), and from organic nitrogen ($R_{ON}$) to quantify PON concentrations.

$$X_{PON} = \frac{(R_{meas} - R_{Cal})(1 + R_{ON})}{(R_{ON} - R_{cal})(1 + R_{meas})} \tag{5}$$

Where ratios from ammonium nitrate calibrations $R_{cal} = 0.5$; $R_{meas}$ = m/z 46:30 ratio from measurements; m/z 46:30 ratio from ON $R_{ON} = 0.1$, Following Kostenidou et al. (2015) consideration, $R_{ON} = 0.1$ was calculated as the minimum m/z 46:30 ratio observed. $R_{ON}$ value of 0.1 has been used in previous studies (Kiendler-Scharr et al., 2016;Tiitta et al., 2016).

$$PON = X_{PON} * NO_3^- \tag{6}$$

Finally, equation 6 calculates PON concentrations [$\mu g.m^{-3}$] where $NO_3^-$ is the total nitrate measured by the cToF-AMS. The

method proposed by Farmer et al. (2010) is based on HR-ToF-AMS measurements were m/z 30 represents $NO^+$ ion and m/z 46 $NO_2^+$ ion while the cToF-AMS gives unit mass resolution mass spectra information, hence, there is the possibility to have interference of $CH_2O^+$ ion at m/z 30. However, when analysing mass spectra from previous laboratory and ambient studies using HR-ToF-AMS to investigate biomass burning emissions, we can confirm that the signal of $CH_2O^+$ at m/z 30 is low compared to signals at m/z's 29 and 31, while in this study m/z 30 is the main signal (Fig. 5.c). Hence, in this study an

interference of $CH_2O^+$ at m/z 30 is unlikely and if there were any interference of $CH_2O^+$ it would be negligible. Table S1 in supplement shows m/z 30/29 and 30/31 from previous laboratory and ambient studies investigating biomass burning emissions.

Another possible interference would be the presence of mineral nitrates at m/z 30 (e.g. $KNO_3$ and $NaNO_3$). However, mineral nitrate salts tend to be large particles (Allan et al., 2006;Chakraborty et al., 2016) and also have low vaporisation

efficiency (Drewnick et al., 2015), which makes it unlikely to be measured by the AMS in large quantities.

### 2.2.3 Multilinear engine 2 (ME-2)

Multilinear engine 2 (Paatero, 1999) is a multivariate solver used to determine factors governing the behaviour of a two dimensional data matrix, which then can be interpreted as pollutant sources. ME-2 uses the same data model as positive matrix factorisation, which is also a receptor model that performs factorisation by using a weighted least squares approach (Paatero and Tapper, 1994).

In order to explore the solution space, ME-2 is capable of using information from previous studies, for example pollutant time series or mass spectra, as inputs to the model (named target time series and target profiles respectively) to constrain the runs. These constraints are performed using the a-value approach, to determine the extent to which the output is allowed to vary. For example, by using an a-value of 0.1 to a specific source, the user is allowing the output to vary 10% from the input. For more details refer to Canonaco et al. (2013).

In this study, ME-2 and PMF were used through the source finder interface, SoFi version 4.8 (Canonaco et al., 2013) to identify OA sources using the suggestions made by Crippa et al. (2014) and the strategy proposed by Reyes-Villegas et al. (2016). ME-2 was performed using mass spectra (BBOA, HOA and COA) from two different studies as target profiles (TP) to constrain the runs; London (Young et al., 2015) and Paris (Crippa et al., 2013), Figure S5 explains the labelling used to identify the different runs.

Solutions were explored with PMF using different fpeak values (ranging from -1.0 to 1.0 with steps of 0.1) and ME-2 using different a-values, (nine runs with London TP and nine runs with Paris TP), looking at four, five and six-factor solutions. Section S7.1 shows the strategy used to determine the optimal solution. Factorisation struggles to separate two or more sources if they are highly correlated, for example during stagnant conditions due to low temperatures and wind speed, which was the case during the bonfire night 2014. The pollutants were well-mixed, making it difficult to separate the sources. Hence, four tests were performed using different time sets in order to identify the best way to perform source apportionment:

- Test 1 performs factorisation on all the dataset.
- Test 2 involves factorising the event before and after bonfire night and using mass spectra from this analysis as TP to factorise the bonfire night event.
- Test 3 involves factorising the bonfire night event and using mass spectra from this analysis as TP as applied to the complete dataset.
- Test 4 involves factorising the event before and after bonfire night and using mass spectra from this analysis as TP to factorise the full dataset.

PON may exhibit covariance with other types of OA, thus their inclusion in the source apportionment analysis may give a more complete factorisation and highlight their co-emission with other OA types. Previous studies have quantified PON concentrations from AMS-PMF analysis to both rural and urban measurements (Sun et al., 2012;Hao et al., 2014;Xu et al., 2015;Zhang et al., 2016). In this study, an experiment was designed by modifying the fragmentation table, through the AMS analysis toolkit 1.56, in order to identify a PON source. The fragmentation table contains the different chemical species measured by the AMS, with each row representing m/z for specific species and where the user can define peaks that exist in each species' partial mass spectrum and their dependency on other peaks (Allan et al., 2004). The following steps were performed to modify the fragmentation table:

- Time series of a new ratio named $R_{ON\_30}$ is calculated by $R_{ON\_30}$ = PON/mz30, were PON is the time series calculated in section 2.2.2 and mz30 is the time series of the signal at m/z=30 measured by the AMS.

- Using the AMS analysis toolkit; the fragmentation table is modified, in the column frag_Organic at the m/z 30, by multiplying $R_{ON\_30}*30$. See figure S4 in supplement for a screenshot of the fragmentation table.

- PMF inputs are generated to be used in SoFi software.

## 3. Results

### 3.1 Meteorology and pollutant overview

During bonfire night festivities on November 5th, a temperature of 4 ˚C and wind speed of 1.5 m.s⁻¹ were observed (Fig. 1.a), causing stagnant conditions which facilitated pollutant accumulation. Looking at the time series for the whole sampling time (Fig. 1.b), it was possible to observe four separate events with different pollutant behaviour (marked with coloured lines over the x-axis in Fig. 1), driven by different meteorological conditions: one event had high secondary concentrations (HSC, yellow line) from October 30th to November 1st, which experienced a relatively high temperature of 17-20 ˚C; one event of low pollutant concentrations (LC, grey line) from November 1st - 3rd was observed when continental air masses were present; Bonfire night (bfo, blue line), with a temperature of 4 ˚C; and a winter-like episode (WL, purple line) from November 8th - 10th, with temperatures of 5-6 ˚C and high primary pollutant concentrations. Figure S3 in supplement shows backtrajectories of the different events.

Aerosol concentrations during bonfire night were particularly high (Fig. 1.c), with the highest peak concentrations of 65.0, 19.0, 6.8, 6.0, 5.9 and 3.2 µg.m⁻³ for OA, BC, $SO_4$, Cl, $NH_4$ and $NO_3$ respectively measured around 20:30 hrs on November 5th. It is worth noting how high these concentrations are compared to concentrations before and after bonfire night (Fig. 1.b) where aerosol concentrations ranged from 0.5 – 7.0 µg.m⁻³. Measured $PM_1$ concentrations (sum of BC, organic and inorganic aerosols) of 115 µg.m⁻³ (Fig. 1.c) were observed during bonfire night.

Looking at the daily concentrations (Fig. 1.d), it is possible to observe $PM_1$ daily concentrations of 25 µg.m⁻³ on bonfire night compared to the low concentrations observed between November 1st – 2nd with concentrations ranging between 3-4 µg.m⁻³. The impact of the emissions during bonfire night is present even during the next day with $PM_1$ concentrations of 14 µg.m⁻³.

Gas phase pollutants were measured at the Whitworth observatory. Figure 2 shows high $SO_2$, CO and $NO_x$ concentrations during bonfire night; these primary pollutants are well known to be combustion related pollutants. The high $SO_2$ concentrations during bonfire night are expected as solid fuels such as wood emit $SO_2$ when burned. This can also explain the $SO_2$ peak in the night of November 10th - 11th when $SO_2$ concentrations may be related to solid fuels used for domestic heating as a result of the low temperatures (6 ˚C). CO and NO were present at higher concentrations during bonfire night compared to previous days with concentrations reaching 1600 ppb (CO) 99 ppb (NO) during bonfire night compared to November 1st with concentrations of 230 ppb of CO and 16 ppb of NO. Some $O_3$ concentrations were measured during bonfire night, but given the very high NO concentrations, these are considered to be an interference with the measurement.

### 3.2 Bonfire night analysis

#### 3.2.1 Traffic and wood burning contributions to BC

OA concentrations started increasing at 19:30 hrs while BC concentrations started increasing two hours earlier around 17:00 hrs (Fig. 1.c). This rise in BC concentrations may be due to bonfire emissions, although they may also be related to traffic emissions; thus the Aethalometer model was used to identify both traffic and wood burning contributions to BC.

Once, $b_{abs}$ are corrected, equations shown in section 2.2.1 are used to apply the Aethalometer model, with Ångström absorption exponents (α) of 1.0 for traffic ($α_{tr}$), using the wavelength 470 nm, and 2.0 for wood burning ($α_{wb}$) using the wavelength 950 nm, to determine traffic and wood burning contributions. Figure 3 shows the absorption coefficients for wood burning $b_{abs\_470wb}$ (blue) and traffic $b_{abs\_950tr}$ (red), both increasing around 17:00-18:00 hrs to values lower than 100 Mm$^{-1}$ while $b_{abs}$, indicating contributions from wood burning and traffic during this event. It is when the majority of bonfire

events are taking place, around 20:00, when $b_{abs\_470wb}$ shows the greatest increase, with values reaching 480 Mm$^{-1}$ compared to 150 Mm$^{-1}$ for $b_{abs\_950tr}$.

### 3.2.2 PON identification and quantification

Currently, there is not a direct technique to quantify on-line integrated PON concentrations. However, it is possible to estimate PON concentrations from AMS measurements using the m/z 46:30 ratios (Farmer et al., 2010) as explained in

section 2.2.2. This event during bonfire night 2014, with high pollutant concentrations, provided the opportunity to identify the presence of PON. Inorganic nitrate from $NH_4NO_3$ has been detected at m/z 46:30 ratios between 0.33 and 0.5 (Alfarra et al., 2006) and of 0.37 (Fry et al., 2009), although each instrument-specific ratio is determined during routine calibrations. PON has been identified with m/z 46:30 ratios of 0.07-0.10 (Hao et al., 2014) and 0.17-0.26 (Sato et al., 2010). In this study, m/z 46:30 ratios of 0.11-0.18 were observed during bonfire night (Fig. 4), confirming the presence of PON during this event.

Figure 4 shows PON concentrations of up to 2.8 µg.m$^{-3}$ during bonfire night, which are over the detection limit of 0.1 µg.m$^{-3}$ reported by Bruns et al. (2010). PON concentrations are considered high compared to previous studies with concentrations between 0.03 – 1.2 µg.m$^{-3}$, from a wide variety of sites across Europe (Kiendler-Scharr et al., 2016), while high PON concentrations of 4.2 µg.m$^{-3}$ were observed during a biomass burning event in Beijing, China (Zhang et al., 2016).

### 3.3 OA source apportionment

This event with high pollutant concentrations during bonfire night gave the opportunity to test ME-2 factorisation tool under these conditions and determine the best way to perform OA source apportionment on a case study event such as this. A number of different approaches to determining the optimal apportionment were tried and the one that yielded the most statistically optimal version treated as a 'best estimate', although it is acknowledged that even this may not be perfect. Indeed, it may not be possible to describe this data completely using the PMF data model. Six different tests were compared;

four tests before modifying the fragmentation table and two tests when modifying the fragmentation table to determine a PON source. Test2_ON resulted to be the optimal 'best estimate' solution, a brief description is given here after being compared to the other tests (Section S7.2 in supplement). From this analysis, test2 resulted to be the best way to deconvolve OA sources, with the lowest parameters analysed: residuals, Q/Qexp values and Chi square. After modifying the fragmentation table, Test2_ON still shows a good performance with low parameters (Fig. S6-S8). Refer to section S7 in

supplementary material for detailed information about source apportionment strategy and analysis performed to determine the optimal solution.

Two steps were involved in Test2_ON: in step a, PMF/ME-2 were run for the event before and after the bonfire night (named as not bonfire event, nbf). In Step b, mass spectra from the solution identified in step a were used as TP, to analyse the bonfire-only (bfo) event. Finally, both solutions (nbf and bfo) were merged for further analysis. Different OA sources

were identified in Test2_ON (Fig. 5), five sources were identified during nbf event: biomass burning OA (BBOA), hydrocarbon-like OA (HOA), cooking OA (COA), secondary particulate organic oxides of nitrogen (sPON_ME2) and low volatility OA (LVOOA). These sources are identified by characteristic peaks at their mass spectrum; BBOA, which is generated during the combustion of biomass, has a peak at m/z 60, related to levoglucosan (Alfarra et al., 2007); HOA, related to traffic emissions, presents high signals at m/z 55 and m/z 57 typical of aliphatic hydrocarbons (Canagaratna et al.,

2004); COA, emitted from food cooking activities, is similar to HOA with a higher m/z 55 and lower m/z 57 (Allan et al., 2010;Slowik et al., 2010;Mohr et al., 2012); LVOOA, identified as a secondary organic aerosol, has a high signal at m/z 44 dominated by the $CO_2^+$ ion (Ng et al., 2010); sPON_ME2 has a strong signal at m/z 30 and it has been identified to be secondary as it follows the same trend as LVOOA (Figure 5.a). In the case of the bfo event six different sources were identified: BBOA, HOA, COA, LVOOA and two factors with peaks at m/z30, which is related to PON (Sun et al., 2012).

These two PON factors may have different sources; one may be secondary (sPON_ME2) and the other primary (pPON_ME2) which has similar trend as BBOA (Fig. 5.b). Further details about pPON_ME2 and sPON_ME2 nature will be explored in section 4.2.

## 4. Discussion

### 4.1 OA source apportionment during bfo event.

It is worth noting that, while all sources have their characteristic peaks and no apparent mass spectral 'mixing' between sources (for example COA with signal at m/z 60), COA, HOA and LVOOA present high concentrations during the bonfire night (Fig. 5.b). High concentrations of these sources could expected as these (traffic and cooking activities) increase before and after the main bonfire events and the night represented a very strong inversion (which will trap all pollutants), but given the high concentrations experienced during the event and known variability for biomass burning emissions, the 'model error'

and thus rotational freedom is likely to be substantial. The result is that these two factors could contain indeterminate contributions from minor variabilities within the biomass burning profile and therefore must be interpreted with caution.

$b_{abs\_470wb}$ has the same source as BBOA, thus the correlation between these two can be used to evaluate the effectivenes of BBOA deconvolution from OA concentrations (Frohlich et al., 2015;Visser et al., 2015). Here $r^2$ values are calculated for the bfo event between $b_{abs\_470wb}$ and the two BBOA's obtained; BBOA, obtained without modifying the fragmentation table and

BBOA_2 obtained after modifying the fragmentation table to identify a PON factor. A higher correlation between $b_{abs\_470wb}$ and BBOA_2 was observed with $r^2 = 0.880$ compared to $r^2 = 0.839$ for $b_{abs\_470wb}$ and BBOA. This improvement on the BBOA_2 is explained by the fact that the PON factor may be mixed with BBOA and when both sources are separated a higher correlation between BBOA_2 and $b_{abs\_470wb}$ is present. There is the possibility the lower r2 between $b_{abs\_470wb}$ and BBOA is due to having two BBOA factors in test2. However, an r2=0.813 between $b_{abs\_470wb}$ and the sum of

BBOA+BBOA_1 is still lower than 0.880.

Here is shown the importance of performing OA source apportionment using different approaches in order to identify the best way to deconvolve OA sources. PMF and ME-2 source apportionment tools could not completely deconvolve OA sources during the bfo event. However, due to the high correlation between for $b_{abs\_470wb}$ and BBOA_2 ($r^2 = 0.880$) we consider that while BBOA_2 might not represent the total OA concentrations from the bonfire night event, it does represent

the trend of OA emmited from the biomass burning.

### 4.2 PON Primary/secondary.

PON concentrations, obtained from the m/z ratios 46:30 (blue line in Fig. 6) have similar trend as BBOA, both increasing at the same time, suggesting a primary origin, but after 22:00 hrs, when BBOA concentrations drop, PON concentrations remain present with a slow decrease and maintaining low concentrations when BBOA concentrations were not present any

more. This suggests the hypothesis that there might not be only one type of PON, and it could be divided into primary and secondary organic nitrate as reported in previous studies performed in western Europe (Mohr et al., 2013;Kiendler-Scharr et al., 2016).

Using this working hypothesis, primary and secondary PON concentrations were estimated using the slope between PON and BBOA, calculated from 18:00 – 12:00 hrs, time when the main bonfire night event took place (Fig. S10). PON concentrations were multiplied by this slope in order to calculate the primary organic nitrate (pPON) and secondary organic nitrate concentrations were calculated as: sPON = PON - pPON. Figure 6 shows the time series of this estimation where pPON reaches 2.5 ugm-3 and sPON with concentrations of 0.5 $\mu g.m^{-3}$.

A similar behaviour with two different PON sources was observed in the source apportionment analysis performed in section 3.3, where it was possible to separate two factors with a peak at m/z 30, characteristic of PON. Figure 7 shows that around 02:00 hrs, concentrations of the pPON_ME2 started to decrease (green line) while sPON_ME2 concentrations (grey line) increased. This analysis shows the presence of two different types of PON, pPON_ME2 is primarily emitted along with BBOA concentrations with the further presence of a different PON, considered to be secondary, due to its increase when primary pollutants start to decrease. Primary and secondary sources of PON have been previously identified from AMS-PMF analysis; Hao et al. (2014) identified PON to be secondary of nature, produced from the interaction between forest and urban emissions, while Zhang et al. (2016) determined PON to be related to primary combustion sources. In this study, is worth noticing that the increase of sPON_ME2 takes places around 2:00 hrs, a period when NO concentrations started decreasing and CIMS-measured $N_2O_5$ and $ClNO_2$ started to increase suggesting that nitrate radical chemistry was occurring (Fig. 8), which is possibly the source of the sPON, although the exact mechanism can only be speculated on.

This nitrogen chemistry process can be observed also in the gas phase, during the day, the main oxidants are $O_3$ and OH while the nitrate radical is unable to participate in daytime chemistry due to being rapidly photolyzed or reacting with NO to produce $NO_2$, (Ayres et al., 2015). Nitrate chemistry at night is important as nitrate radicals can be the main oxidants in polluted nocturnal environments away from enhanced NO and can create reservoirs and sinks of $NO_x$. The main $NO_x$ removal at night is via the uptake of dinitrogen pentoxide ($N_2O_5$) into aerosols, as at night $N_2O_5$ is formed from $NO_3$ and $NO_2$. In the presence of chloride in the particle phase (e.g. in sea salt particles), $N_2O_5$ reacts to produce nitryl chloride ($ClNO_2$). In the morning, following overnight accumulation of $ClNO_2$, photochemical reactions take place to produce Cl and $NO_2$. $N_2O_5$ and $ClNO_2$ processing an interactions with nitrate chemistry has been previously studied in the UK (Le Breton et al., 2014a;Bannan et al., 2015). Figure 8 shows $N_2O_5$, $ClNO_2$ and $O_3$ concentrations increasing when NO and $NO_2$ concentrations decrease. All these processes may facilitate the sPON production at night. $N_2O_5$ concentrations reduce quickly after the sun rises, around 08:00 hours, while $ClNO_2$ concentrations decrease at a slower rate, with the lowest concentrations observed around 13:00 hours. Along with $NO_3$ chemistry, it was possible to observe other nitrogen containing gases during bonfire night using the CIMS such as hydrogen cyanide (HCN) and Nitrous acid (HONO), which have been found to be emitted from fires (Le Breton et al., 2013;Wang et al., 2015). High HONO concentrations at night are important during next morning when HONO reacts to produce OH and NO which impacts both the OH budget and $NO_x$ concentrations early next morning (Lee et al., 2016).

**4.3 OA factors and CIMS correlations**

Analysing the CIMS measurements and comparing them with the OA factors, it may be possible to identify gas markers that can be used as inputs (target time series) to constrain solutions in future ME-2 analyses or as proxies when AMS data is not available. A linear regression was performed between the OA sources determined in section 3.4.1 and CIMS peaks that have been considered positively identified (Priestley et al., in preparation), performing a coefficient of determination ($r^2$) analysis for the complete dataset (ALL), and the event HSC, LC, bfo and WL. During the event HSC, none of the OA sources showed an $r^2$ higher than 0.6. HOA did not have an $r^2$ higher than 0.6 with any of the different events analysed. There were no specific markers identified for COA, while COA showed $r^2$ values higher than 0.6 for bfo event, these $r^2$'s were observed also with BBOA even with higher values. Table S4 shows the $r^2$ values, higher or equal to 0.4, obtained in this analysis. It is

worth noting that $r^2$ values in ALL event seem to be influenced by the bfo event; this is the case for BBOA, COA and LVOOA which show similar $r^2$ values in both events. Thus, the analysis will be explained only in the individual events: bfo LC, and WL.

As expected, during bfo, BBOA is the OA source that shows the highest number of correlations during bonfire night. During bfo episode, high correlations ($r^2$) were observed with BBOA and methacrylic acid (0.92), Acrylic acid (0.90), nitrous acid (0.86), propionic acid, (0.85) and Hydrogen cyanide (0.76), which have been previously determined as biomass burning tracers (Veres et al., 2010;Le Breton et al., 2013). Formic acid presented a high correlation (r2=0.86) with BBOA during bonfire night, however this value drops to 0.52 for all the dataset, which suggests formic acid during bonfire night is mainly primary while formic acid concentrations measured for the whole dataset may be related to primary and secondary sources. This agrees with Le Breton et al. (2014b) who explored both primary and secondary origins of formic acid.

During the bfo event, LVOOA did not show a characteristic gas marker as all the $r^2$ values were also observed with BBOA. This suggests two hypotheses, that the LVOOA was mixed with BBOA, in the form of humic-like material (Paglione et al., 2014), which cannot be differentiated from secondary OA in the mass spectra (Fig. 5.c), or it could also be that secondary LVOOA may actually be present at the same time as BBOA concentrations, as it has been observed that during high relative humidity and low temperature enhanced partitioning of semi-volatile material to the particle phase occurs, where subsequent oxidation and oligomerisation may occur. Moreover, due to the high aerosol concentration present during the bonfire night, there is more surface available for gases to be condensed and more particulate bulk to absorb into, thus it could be speculated that there would be high secondary aerosol concentrations. However, this is deemed unlikely, as there is unlikely to be much gas phase oxidation occurring in the presence of such high NO concentrations, which will remove ozone and nitrate radicals, the main source of oxidants at night.

During the bfo event, pPON_ME2 showed high $r^2$ values with carbon monoxide (0.78) as well as hydrogen cyanide (0.77), Methylformamide (0.65) and Dimethylformamide (0.63) which are typical primary pollutants related to combustion processes [(Borduas et al., 2015) and references therein]. sPON_ME2 showed low correlations with ClNO$_2$ (0.52) and ClNO$_3$ (0.53). High $r^2$ values were also observed during LC episode between ClNO$_2$ - ClNO$_3$ and LVOOA (0.67 - 0.66) and sPON (0.74 - 0.69) proving their secondary origin. Cl$_2$, which has been previously identified to be related to both primary and secondary sources (Faxon et al., 2015), shows low correlations with pPON_ME2 (0.44) during bfo event and sPON_ME2 (0.55) during LC event.

### 4.4 PON and its relationship with b$_{abs\_470wb}$ and BBOA

Organic oxides of nitrogen, originating from biomass burning, have been previously found to absorb light near the UV region (Jacobson, 1999;Flowers et al., 2010;Mohr et al., 2013). However, there is still a question of whether this absorption is due to primary or secondary PON. Here, the relationship between $b_{abs\_470wb}$ and PON and BBOA will be analysed to determine if sPON absorbs at 470 Mm$^{-1}$, which would interfere with Aethalometer measurements.

In order to quantitatively determine any contribution from organic nitrates to the Aethalometer data products, a multilinear regression (MLR) analysis was performed on the complete dataset dataset (ALL), and the events HSC, LC, bfo and WL (Table 1). This analysis was done in three ways: MLR 1 with BBOA from OA source apportionment without modifying the FragPanel and PON from m/z 46:30 analysis; MLR 2 with BBOA_2 from OA source apportionment after modifying the FragPanel and PON from 46:30 analysis; MLR 3 with with BBOA_2 and PON sources from OA source apportionment after modifying the FragPanel. The following bilinear regression was used:

$b_{abs\_470wb}$ = A+ B*x1+C*x2 (6)

When the parameter s"D" and x3 are used it means a trilinear regression was performed. MLR1 x1=BBOA, x2=PON. MLR2 x1=BBOA_2, x2=PON. MLR3 x1= BBOA_2, x2=sPON_ME2, x3=LVOOA in HSC and x3= pPON in bfo. A is the origin and the partial slopes B, C and D represent the contribution of x1 ,x2 and x3 to $b_{abs\_470wb}$, respectively.

As used in previous studies (Elser et al., 2016;Reyes-Villegas et al., 2016), multilinear regression analysis allows to determine the relationship of one parameter between two or more variables. Here we are analysing the partial slopes and origin to determine the correlation of $b_{abs\_470wb}$ with the other variables. Table 1 shows the MLR outputs where; A represents the background, B, C and D represent the partial slope between $b_{abs\_470wb}$ and the respective organic aerosol. B/C represents the ratio between B and C partial slopes, with the following considerations: if B/C < 1 then, there is a higher contribution of organic nitrates to $b_{abs\_470wb}$; if B/C > 1 then, there is a higher contribution of BBOA to $b_{abs\_470wb}$. Looking at the coefficient of determination of the multilinear regression ($r^2\_MLR$) for the three MLR analyses, it is possible to observe that HSC and LC events present low $r^2\_MLR$ values ranging from 0.064 and 0.480, while bfo and WL events have high correlations with values between 0.760 and 0.910, which shows that is during high primary OA emissions are present when a high correlation between $b_{abs\_470wb}$ and BBOA and PON is observeed.

These high $r^2$ values, particularly during the bfo event which presented the highest $r^2$ (0.910), are consistent with previous studies that found organic nitrates absorb at short wavelengths; Mohr et al. (2013) identified correlation values of 0.65 between nitrophenols and $babs_{370wb}$. Teich et al. (2017), in a recent study from offline filters, determined nitrated aerosol concentrations with further analysis of the light absorption of aqueous filter extracts ($babs_{370}$), identified $r^2$ values between $b_{abs\_370}$ and nitrated aerosol concentrations of 0.67 to 0.74 depending on acidic or alkaline conditions respectively.

In MLR 3, it is possible to observe that, during bfo event, the main contribution to $b_{abs\_470wb}$ is attribuited to both BBOA_2 (16.657) and pPON_ME2 (7.357) while babs:sPON_ME-2 values were zero, with an optimum $r^2$ of 0.910. This lack of correlation between babs and sPON is observed in the linear regression babs:sPON_ME2 with an $r^2$ of 0.188. These results show that while there is evidence of pPON_ME2 absorbing at 470 nm, with a partial slope of 16.657, sPON_ME2 did not show to be absorbing at 470 nm. The implication of the background not going to zero (6.093) is that there is still an unexplained contribution to the absorbtion at 470 nm, unrelated to sPON_ME2.

In order to further explore the possibility of sPON_ME-2 absorbing at 470 nm, the HSC event was analysed, where sPON_ME2 was shown to be non-absorbing at 470 nm with a partial slope of zero. On the other hand, BBOA_2 had a partial slope of 27.288 and bkgd a value of 2.527. This background value suggests there is another component related to $b_{abs\_470wb}$ that is not sPON. Thus, a trilinear regression was performed to *HSC between $b_{abs\_470wb}$ and BBOA_2, sPON and LVOOA. Here, the background value drops to 1.649, sPON partial slope is zero and LVOOA presents a partial slope of 1.138. These results confirm that sPON do not absorb light at 470 nm while LVOOA, or at least part of the components of LVOOA, absorb during the HSC event and pPON_ME2 absorbs during the bfo event.

These results agree with previous studies that found biomass burning OA contain important concentrations of light absorbing brown carbon (BrC) and that certain types of SOA are effective absorbers near UV light (Bones et al., 2010;Saleh et al., 2014;Washenfelder et al., 2015). The fact that pPON_ME2 and LVOOA were shown to be absorbing light at a short wavelength (470 nm) will have a direct impact on Aethalometer model studies; while pPON_ME2 could be considered a component of the wood burning aerosol apportioned using the Aethalometer, it may be that there is an interference from other forms of BrC in SOA. However, this work would suggest that sPON specifically does not contribute to the latter, so a different component of LV-OOA would have to be responsible. As well as Aethalometer interpretation, it is also worth mentioning that these findings may have implications for studies on the radiative properties of the atmosphere, as BrC is also thought to affect climate (Jacobson, 2014).

## 5. Conclusions

In order to better understand the aerosol chemical composition and variation in source contribution during periods of nocturnal pollution, online measurements of gases and aerosols were made in ambient air between 29th October and 10th November 2014, at the University of Manchester with detailed analysis of the special high pollutant concentrations during bonfire night celebrations on 5th November. High aerosol concentrations were observed during the bonfire night event with 115 µg.m$^{-3}$ of PM$_1$. Important nitrogen chemistry was present with high HCN, HCNO and HONO concentrations primarily emitted with further presence of N$_2$O$_5$ and ClNO$_2$ concentrations from nocturnal nitrate chemistry taking place after NO$_x$ concentrations decreased.

Organic aerosol source apportionment was performed using the ME-2 factorisation tool. The particular high pollutant concentrations together with the complex mix of emissions did not allow the running of ME-2 for the complete dataset, thus the dataset was divided into different event s. The best way to perform source apportionment was found to be to, (a) analyse the event before and after the bonfire night using BBOA, HOA and COA from a previous study in Paris as TP, and (b) conduct a further ME-2 analysis of the bonfire night event using BBOA, HOA and COA mass spectra from (a) as TP. Moreover, an improvement on the source apportionment was observed after modifying the fragPanel in order to identify organic nitrate sources, increasing the r$^2$ value from linear regressions between $b_{abs\_470wb}$ (absorption coefficient of wood burning at 470 nm) and BBOA from 0.839 to 0.880. PMF and ME-2 source apportionment tools could not completely deconvolve OA sources during the bfo event as LV-OOA, COA and HOA may be mixed with BBOA concentrations. However, due to the high correlation between for $b_{abs\_470wb}$ and BBOA (r$^2$ = 0. 880) we consider that while BBOA might not represent the total OA concentrations from the bonfire night event, it does represent the trend of OA emmited from the biomass burning.

The combination of CIMS measurements and OA sources determined from AMS measurements provided important information about gas tracers to be used as inputs (target time series) to improve future ME-2 analyses, particularly gases correlating with BBOA, LVOOA and secondary particulate organic nitrate. However the use of these species as target time series should be used with care as their time variation are greatly affected by meteorological conditions.

The presence of two classes of particulate organic nitrate (PON), secondary (sPON_ME2) and primary (pPON_ME2) PON, was identified both from looking at the BBOA:PON relationship and from the ME-2 analysis after modifying the FragPanel. It is clear that, during bonfire night, pPON_ME2 concentrations increased when BBOA concentrations are present and sPON_ME2 concentrations started evolving when the primary concentrations decreased.

It was determined that pPON_ME2 absorbed light at a wavelenght of 470 nm during the bonfire night, where the multilinear regression perfomed between $b_{abs\_470wb}$, BBOA and pPON_ME2 showed a high r$^2$ of 0.910 while sPON_ME2 did not contribute to light absorption at 470nm. During the HSC episode, LVOOA showed a partial slope of 1.138 in the multilinear regression and an r$^2$ from linear regression with $b_{abs\_470wb}$ of 0.225, implying secondary LVOOA (associated with SOA) may be absorbing at 470 nm and sPON_ME2 was not absorbing at this wavelength. These results will help us to understand the mechanistic contributions to UV absorption in the Aethalometer and will have direct implications for source apportionment studies, which may need to be corrected for secondary organic aerosol interferences near the UV region.

## 6. Data availability

Processed data is available through the archive at the British Atmospheric Data Centre (http://badc.nerc.ac.uk/browse/badc), with search term `COMPART'. Raw data is archived at the University of Manchester and is available on request.

*Author contributions.* Ernesto Reyes-Villegas, Michael Flynn, Hugh Coe, Carl Percival, James Allan designed the project; Ernesto Reyes-Villegas, Yu-Chieh Ting, Sophie Haslett, Thomas Bannan, Michael Le breton, Paul Williams,  Asan Bacak, operated, calibrated and performed QA of instrument measurements; Ernesto Reyes-Villegas and Michael Priestley performed the data analysis; Ernesto Reyes-Villegas, Hugh Coe and James Allan wrote the paper.

*Acknowledgements.* This work was supported through the UK Natural Environment Research Council (NERC) through the Com-Part (grant ref: NE/K014838/1). Ernesto Reyes-Villegas is supported by a studentship by the National Council of Science and Technology-Mexico (CONACYT) under registry number 217687.

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

## Appendix A: Source apportionment solution without modifying the fragmentation table.

Figure A presents results obtained with test2. Figure A.c shows mass spectra of the two chosen solutions; five sources were identified during nbf period: BBOA, HOA, COA, SVOOA and LVOOA. In the case of the bfo period six different sources were identified: BBOA, HOA, COA, factor4 which seems to be a mixed factor with a peak at m/z43 (characteristic of SVOOA) and peaks at m/z 55 and m/z 57 (characteristic of HOA), LVOOA and BBOA_1. BBOA_1 source appears to be mixed between LVOOA (peaks at m/z28 and mz44) and BBOA (peak at m/z60). We can see here, that while test2 resulted to be the best way to deconvolve OA sources compared to tests 1,3 and 4, it still shows mixing with SVOOA, LVOOA, and BBOA_1. Situation that improved when doing OA source apportionment after modifying the fragmentation table in test2_ON.

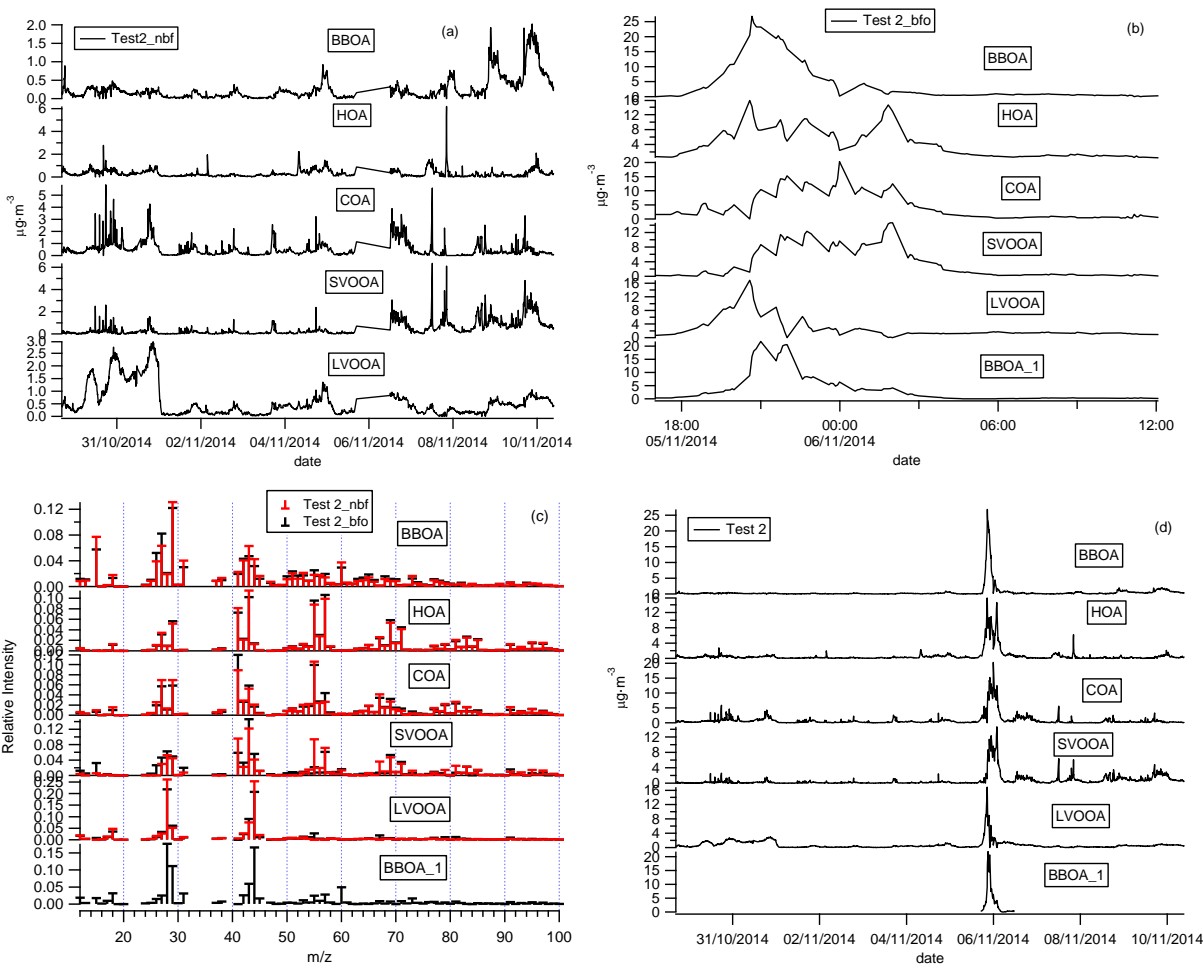

**Figure A: OA sources mass spectra and time series for test 2.**

**Appendix B: Symbols and description of main parameters used.**

| Symbol | Name |
|---|---|
| | **Events** |
| bfo | bonfire-only event (November 05-17:00 hrs – 06 - 12:00 hrs) |
| nbf | not bonfire (before and after bonfire night) |
| HSC | High secondary concentrations (October 30th to November 1st) |
| LC | Low concentrations (November 1st - 3rd ) |
| WL | Winter-like (November 8th -10th) |
| | **Aethalometer correction and model** |
| $\alpha$ | Ångström absorption exponent |
| $\alpha_{tr}$ | Ångström absorption exponent for traffic |
| $\alpha_{wb}$ | Ångström absorption exponent for wood burning |
| $ATN$ | Atennuation |
| BC | black carbon [$\mu g.m^{-3}$] |
| $b_{abs}$ | absorption coefficient [$m^{-1}$] |
| $b_{abs\_470}$ | absorption coefficient at 470 nm [$m^{-1}$] |
| $b_{abs\_950}$ | absorption coefficient at 950 nm [$m^{-1}$] |
| $\sigma_{ATN}$ | attenuation cross section [$m^2.g^{-1}$] |
| $\lambda$ | wavelength  [nm] |
| $b_{ATN}$ | Uncorrected absorption coefficient [$Mm^{-1}$] |
| $b_{abs}$ | Corrected absorption coefficient [$Mm^{-1}$] |
| C | Multiple scattering correction constant |
| R | Filter loading correction |
| $f$ | shadowing factor |
| | **Organic aerosol factors (OA)** |
| BBOA | Biomass burning organic OA obtained without modifying the fragmentation table |
| BBOA_1 | Second biomass burning organic OA obtained without modifying the fragmentation table |
| BBOA_2 | biomass burning organic OA obtained after modifying the fragmentation table |
| HOA | Hydrocarbon-like OA |
| COA | Cooking OA |
| SVOOA | Semivolatile OA |
| LVOOA | Low volatility OA |
| PON | Particulate organic nitrate, calculated with 46:30 ratios. |
| pPON | Primary particulate organic nitrate, estimated using the slope between PON and BBOA |
| sPON | Secondary particulate organic nitrate, sPON = PON - pPON |
| pPON_ME2 | Primary particulate organic nitrate, calculated from ME-2 analysis |
| sPON_ME2 | Secondary particulate organic nitrate, calculated from ME-2 analysis |

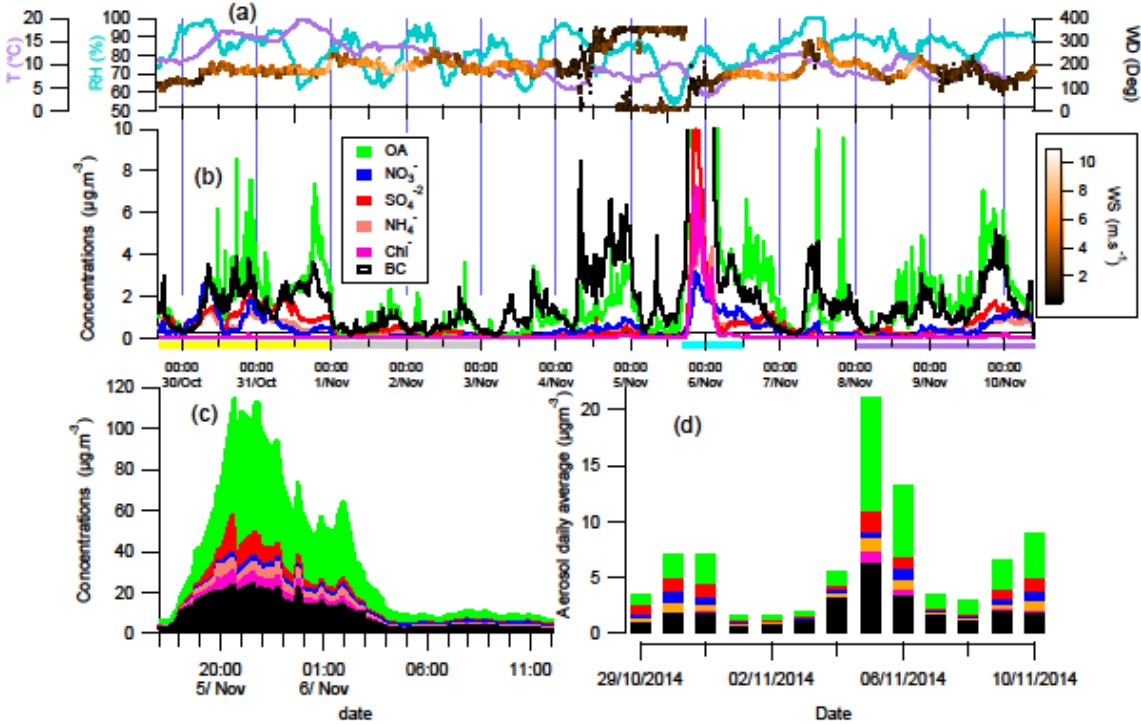

**Figure 1: Meteorology (a), aerosol concentrations during all measurement period (b). Chemical component mass concentrations during bonfire night plotted cumulatively (c). Daily aerosol concentrations (d).**

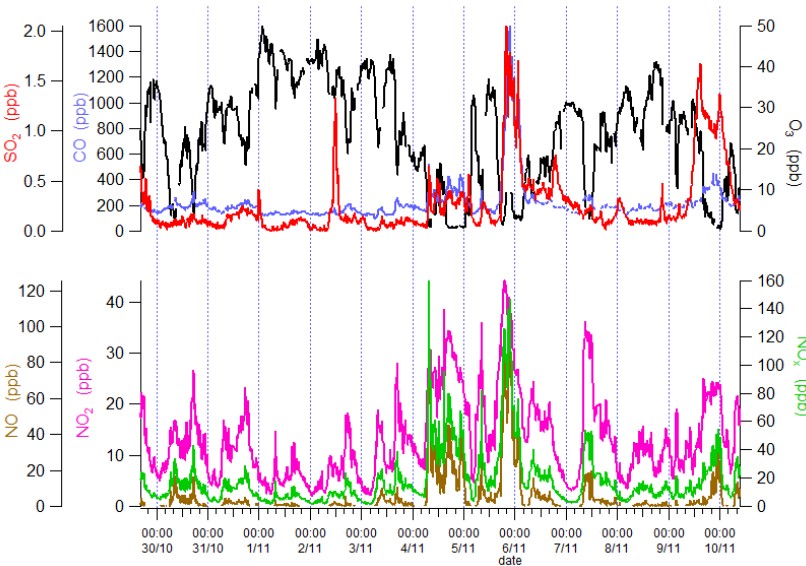

**Figure 2: Time series of gases measured at Whitworth observatory.**

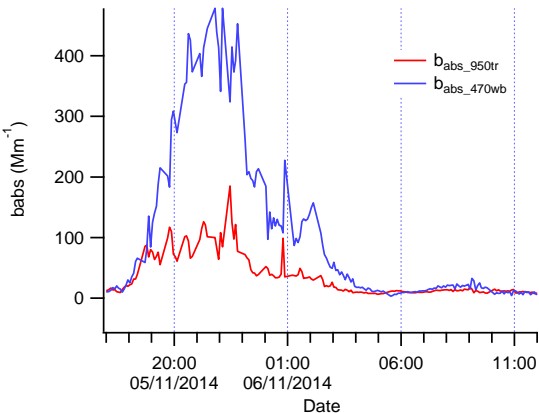

820 **Figure 3: Absorption coefficients for Wood burning (wb) and traffic (tr).**

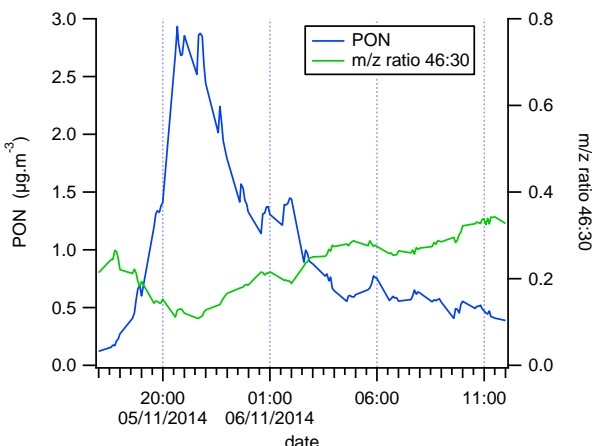

**Figure 4: PON concentrations during bonfire night.**

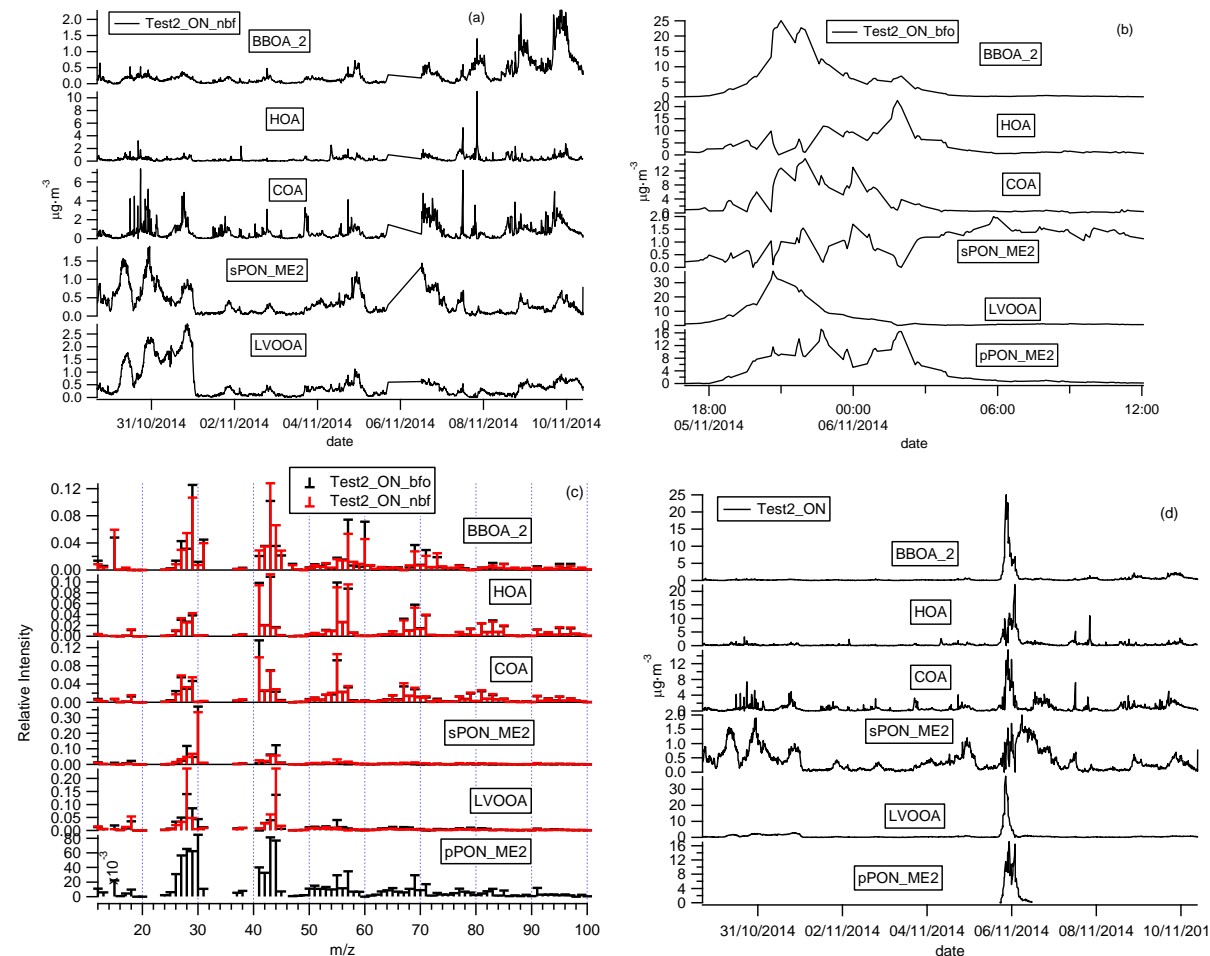

**Figure 5: OA sources mass spectra and time series for test 2_ON for bonfire only (bfo) and not bonfire events (nbf). Figure 6.d shows time series of both events.**

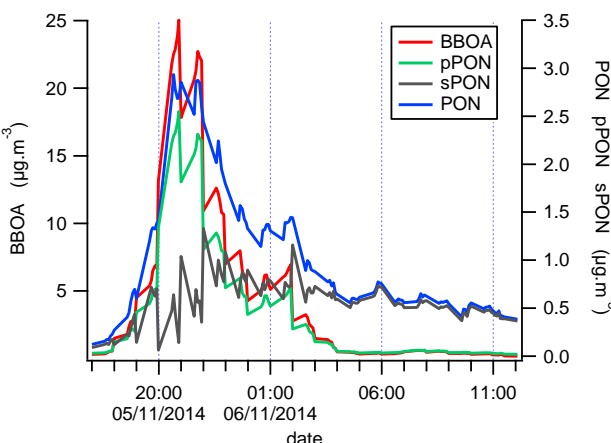

**Figure 6: Secondary (sPON) and primary (pPON) organic nitrate time series estimated from PON and BBOA.**

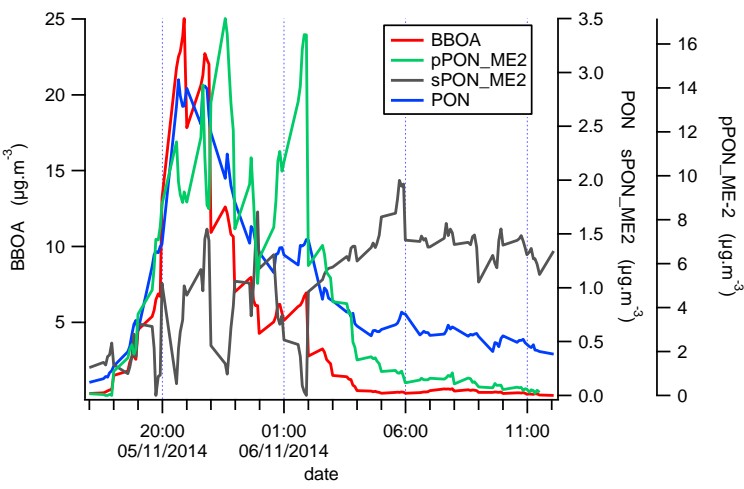

**Figure 7: Secondary and primary organic nitrate time series obtained from ME-2 analysis.**

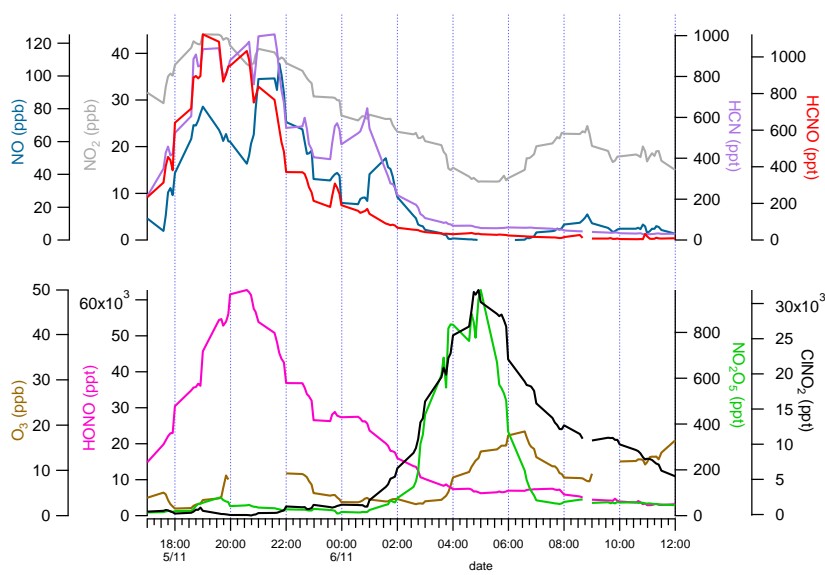

**Figure 8: Time series of gases pollutants during bonfire night.**

**Table 1: Multilinear and linear regression analysis between $b_{abs\_470wb}$ and organic aerosols.**

| | | | All | HSC | LC | bfo | WL |
|---|---|---|---|---|---|---|---|
| **MLR 1** | A | bkgrd | 0.000 | 4.555 | 1.004 | 0.000 | 1.293 |
| | B | babs:BBOA | 14.340 | 3.547 | 18.284 | 11.926 | 10.318 |
| | C | babs:PON | 54.495 | 9.212 | 12.046 | 73.115 | 21.724 |
| | | B/C | 0.263 | 0.385 | 1.518 | 0.163 | 0.475 |
| | | r²_MLR | 0.912 | 0.064 | 0.364 | 0.898 | 0.760 |
| **Linear 1** | r² | babs:BBOA | 0.861 | 0.043 | 0.358 | 0.839 | 0.739 |
| | r² | babs:PON | 0.819 | 0.060 | 0.275 | 0.897 | 0.311 |
| **MLR 2** | A | bkgrd | 0.000 | 2.527 | 0.753 | 0.000 | 0.079 |
| | B | babs:BBOA_2 | 15.653 | 27.288 | 26.481 | 14.319 | 10.018 |
| | C | babs:PON | 42.840 | 0.000 | 1.200 | 54.353 | 18.982 |
| | | B/C | 0.365 | *** | 22.060 | 0.263 | 0.528 |
| | | r²_MLR | 0.922 | 0.392 | 0.480 | 0.902 | 0.804 |
| **Linear 2** | r² | babs:BBOA_2 | 0.894 | 0.392 | 0.480 | 0.880 | 0.788 |
| | r² | babs:PON | 0.819 | 0.060 | 0.275 | 0.897 | 0.311 |

| | | | All | HSC | *HSC | LC | *bfo | WL |
|---|---|---|---|---|---|---|---|---|
| **MLR 3** | A | bkgrd | 0.000 | 2.527 | 1.649 | 0.763 | 6.093 | 0.000 |
| | B | babs:BBOA_2 | 21.545 | 27.288 | 22.764 | 26.668 | 16.657 | 8.577 |
| | C | babs:sPON_ME2 | 3.926 | 0.000 | 0.000 | 0.191 | 0.000 | 9.017 |
| | D | | | | 1.138 | | 7.357 | |
| | | B/C | 5.488 | *** | *** | *** | *** | 0.951 |
| | | B/D | | | 20.005 | | 2.264 | |
| | | r²_MLR | 0.896 | 0.392 | 0.418 | 0.480 | 0.910 | 0.803 |
| **Linear 3** | r² | babs:BBOA_2 | 0.894 | 0.392 | 0.392 | 0.480 | 0.880 | 0.788 |
| | r² | babs:sPON_ME2 | 0.024 | 0.000 | 0.000 | 0.273 | 0.188 | 0.647 |
| | r² | r²_D | | | 0.225 | | 0.633 | |

*Trilinear regression was performed as in *bfo analysis there were two PON factors from ME-2 analysis; pPON and sPON, with D= slope babs:pPON, r²_D = r² babs:pPON. In *HSC analysis; BBOA, sPON and LVOOA were used, with D= slope babs:LVOOA, r²_D = r² babs:LVOOA. PON is the particulate organic nitrate estimate from 46:30 ratios. All= complete dataset; HSC= Episode with high secondary concentrations (October 30th to November 1st), LC = Episode with low concentrations (November 1st - 3rd); bfo = epidose with bonfire only concentrations (05-Nov 17:00 hrs – 06-Nov 12:00 hrs); WL= Episode with winter-like carachteristics (November 8th -10th).