# Peer review of "Simultaneous Aerosol Mass Spectrometry and Chemical Ionisation Mass Spectrometry measurements during a biomass burning event in the UK: Insights into nitrate chemistry"

_Atmospheric Chemistry and Physics, 2017_

## Referee Comment (RC1) · Anonymous Referee #1 · 11 Aug 2017

Review of "Simultaneous Aerosol Mass Spectrometry and Chemical Ionisation Mass Spectrometry measurements during a biomass burning event in the UK: Insights into nitrate chemistry" by Reyes-Villegas et al.

In this study, the authors use cToF-AMS, I⁻ HR-ToF-CIMS, and Aethalometer to characterize one biomass burning event. The authors identified both primary and secondary particulate organic nitrate (PON) in the biomass burning even. The secondary PON is proposed to arise from $NO_3\bullet$ chemistry. Further, it is found that while primary PON can absorb light at 470nm, the secondary PON does not. The topic is clearly of interest to the community. However, I have some concerns about the PMF analysis, particularly regarding that the concentrations of all factors increase during the biomass burning event. Also, the discussions on the relationship between absorption and different OA components need substantial revisions. For example, all the r2 values in section 4.4 do not match the values in table 1, which makes the discussion difficult to follow. At this time, I recommend accept with major revisions.

Major Comments

1.    PMF results.

It has been challenging to perform source apportionment on a dataset including a special event with high concentrations. This study takes an important step to address this issue. However, the PMF results are still not satisfactory. My major concern is that all OA factors show significant increase during the biomass burning event (Figure 5 and 6). LV-OOA increases by ~30 µg m⁻³. Since fresh biomass burning unlikely contribute to LV-OOA, the increase in LV-OOA suggests PMF artifacts. The COA and HOA increase by ~8 µg m⁻³ and ~20 µg m⁻³ during the biomass burning, respectively. These enhancement magnitudes cannot be explained by the inversion at night. The enhanced concentrations of LV-OOA, COA, and HOA during the biomass burning event are likely interference from biomass burning.

Using the suggested two-step approach, a clean BBOA factor still cannot be resolved. For example, BBOA_1 is a mixed factor between LV-OOA and BBOA. Have the authors tried PMF2 solver on the whole dataset? Including the biomass burning event would be useful to get a clear BBOA factor, which helps to identify the BBOA concentration during the non-biomass burning

period. However, the disadvantage of this method is that the concentrations of all OA factors would falsely decrease during the biomass burning.

The authors have done careful evaluation on PMF results. PMF results from two different tests (test 2 and test 2_ON) are presented, but the PMF results are different. This causes many confusions. For example, why does the mass spectrum of BBOA change between test 2 and test 2_ON (i.e., BBOA-2 vs BBOA)? Why are two BBOA factors are resolved in test 2, but only one BBOA factor in test 2_ON? Why is SV-OOA only resolved in test 2, but not in test 2_ON? I suggest the authors to present only the most reasonable/best solution in the manuscript to avoid confusion.

Details about PMF analysis are included details in the SI. However, I suggest that some key points should be briefly discussed in the main text as well. For example, Line 222, it should be justified that why test 2 is the best. What criteria do the authors look at?

2.      Discussions on the light absorbing properties of OA components are confusing and require substantial revisions.

(1) The most important issue is that the r2 values in the manuscript do not match those in table 1.

(2) What's the rationale behind eqn. 6? Why is the partial slope used? In MLR3, there is no sPON_ME2. Then how is the light-absorption of sPON_ME2 evaluated?

(3) In the abstract, it is stated that LV-OOA absorb light at 470nm over that of black carbon. Where is the justification for this conclusion?

(4) Line 278, the authors state that after modifying the fragmentation table, the correlation between $b_{abs\_470wb}$ and BBOA is improved. I wonder if the improvement is mainly because that there are only one BBOA is resolved in test 2_ON, but two BBOA factors in test 2? In other words, is the improved correlation simply due to that modifying the fragmentation table somehow helps to separate the BBOA factor? What's the r2 between $b_{abs\_470wb}$ and the sum of BBOA and BBOA_1 in test 2?

(5) Line 374, the authors need to be cautious that not all organic nitrates can absorb light. Most identified light-absorbing organic nitrate are nitro aromatic compounds (Lin et al., 2016; Mohr et al., 2013).

3.      The separation of primary PON (pPON) and secondary PON (sPON).

(1) How do the authors identify pPON and sPON? In Figure 6, the sPON_ME2 has more evident signal at m/z 30 than pPON. What does the mass spectrum of organics that associate with sPON

look like? Where are the organic signals associated with pPON_ME2 from? Fresh biomass burning? More discussions regarding pPON_ME2 and sPON_ME2 are required.

(2) The authors use two methods to differentiate pPON and sPON. However, there are discrepancies in the results (Figure 7 vs. 8). For example, pPON_ME2 decreases slower than BBOA in Figure 8. Could the authors directly compare the results from these two methods (i.e., scatter plot)?

(3) Previous studies have attempted to run PMF analysis on combined organic and nitrate mass spectra (Sun et al., 2012; Xu et al., 2015). The authors should compare to other literature.

Minor Comments

1.      Line 68. It should be "Ng et al., 2017".

2.      Line 212. What's the $NO_2^+/NO^+$ value of organic nitrate used in this study? This information should be mentioned in the main text.

3.      Line 227. The mass spectrum of factor 4 is very similar to that of SV-OOA in step a. Then how do the authors justify "SV-OOA" in step a?

4.      Line 392, please cite Washenfelder et al. (2015), which showed that biomass burning OA is light absorbing.

Reference

Lin, P., Aiona, P. K., Li, Y., Shiraiwa, M., Laskin, J., Nizkorodov, S. A., and Laskin, A.: Molecular Characterization of Brown Carbon in Biomass Burning Aerosol Particles, Environ Sci Technol, 50, 11815-11824, 10.1021/acs.est.6b03024, 2016.

Mohr, C., Lopez-Hilfiker, F. D., Zotter, P., Prévôt, A. S. H., Xu, L., Ng, N. L., Herndon, S. C., Williams, L. R., Franklin, J. P., Zahniser, M. S., Worsnop, D. R., Knighton, W. B., Aiken, A. C., Gorkowski, K. J., Dubey, M. K., Allan, J. D., and Thornton, J. A.: Contribution of Nitrated Phenols to Wood Burning Brown Carbon Light Absorption in Detling, United Kingdom during Winter Time, Environ Sci Technol, 47, 6316-6324, 10.1021/es400683v, 2013.

Sun, Y. L., Zhang, Q., Schwab, J. J., Yang, T., Ng, N. L., and Demerjian, K. L.: Factor analysis of combined organic and inorganic aerosol mass spectra from high resolution aerosol mass spectrometer measurements, Atmos. Chem. Phys., 12, 8537-8551, 10.5194/acp-12-8537-2012, 2012.

Washenfelder, R. A., Attwood, A. R., Brock, C. A., Guo, H., Xu, L., Weber, R. J., Ng, N. L., Allen, H. M., Ayres, B. R., Baumann, K., Cohen, R. C., Draper, D. C., Duffey, K. C., Edgerton, E., Fry, J. L., Hu, W. W., Jimenez, J. L., Palm, B. B., Romer, P., Stone, E. A., Wooldridge, P. J., and Brown, S. S.: Biomass burning dominates brown carbon absorption in the rural southeastern United States, Geophysical Research Letters, 2014GL062444, 10.1002/2014GL062444, 2015.

Xu, L., Suresh, S., Guo, H., Weber, R. J., and Ng, N. L.: Aerosol characterization over the southeastern United States using high-resolution aerosol mass spectrometry: spatial and seasonal variation of aerosol composition and sources with a focus on organic nitrates, Atmos. Chem. Phys., 15, 7307-7336, 10.5194/acp-15-7307-2015, 2015.

---

## Referee Comment (RC2) · Anonymous Referee #2 · 12 Sep 2017

Review for: Simultaneous Aerosol Mass Spectrometry and Chemical Ionisation Mass Spectrometry measurements during a biomass burning event in the UK: Insights into nitrate chemistry by Reyes-Villegas et al.

This paper describes the measurements of the chemical composition and absorption of aerosols during the bonfire night event in Manchester with particular focus on primary and secondary particulate organic nitrate. While the topic of this paper is interesting to the community, it has some major issues resulting from (1) the use of the C-ToF-AMS and not an HR-AMS for the quantification of PON and (2) the PMF not being able to

resolve the biomass burning event from all the other factors without crosstalk between the different factors. In general, the various steps of the PMF are poorly explained and are very difficult to follow, especially for a "non-AMS reader". (3) There are also various parts of the paper that are redundant and others are missing, for example there were different PMF tests performed, but only one was used. So why describe the first test at all. On the other hand a description of the AMS instrument and quantification or a description of the PMF factors is missing. With all those issue, I cannot recommend this paper for publication in its current form without major improvement to the PMF analysis and results.

Major Comments:

Quantification of PON from biomass burning using a CToF-AMS: The m/z 46:30 ratio is used for the quantification of PON, which is based on a method done by Farmer et al 2010 that used an HR-AMS and did not include any biomass burning emissions. The large fraction of OA from biomass burning certainly will produce a large $CH_2O+$ contribution and will make the quantification of PON very different compared to the Farmer et al paper. On page 7 line 250 the authors say that the m/z 30 interference is likely small, but this might be very different in biomass burning and needs to be shown or given a reference. In addition it is written on page 6 line 206 that the interference of $CH_2O+$ is discussed in section 3.4.1, but no such section exists in the paper. So it is not clear, how the authors deal with the m/z 30 interference and how large the uncertainty or error estimate on the quantification of PON is. In addition in the later section, where the primary versus secondary PON is discussed, it is certainly possible that the m/z 30 interference is different as the composition changes during the later part of the night, when the authors claim that they observed secondary PON.

A section needs to be added, where the interference is clearly explained and the effect needs to be quantified. This should result in an uncertainty range for the m/z 30:46 ratio and error estimate for PON. All of this should be added to the instrumentation section, where there needs to be a description of the AMS added as well that includes

a discussion on the calibration for NH4NO3 and, if available, for PON compounds. Here it should be made clear what m/z 30:46 ratio NH4NO3 has in this particular instrument? The resulting error estimate needs to be taken into account for the following discussions.

For biomass burning the fragmentation tables need to be modified, which the authors do later in the paper. It does not make sense to me to run PMF on not-corrected data as was done in Section 3.3, when you know you are using incorrect data.

The section on the FragPanel modification is very specific to AMS users and all the three bullet points cannot be understood by anybody else without explaining all the abbreviations and acronyms. Especially sentences like on page 7 line 238: time series of PON:mz30 were calculated with the equation PON:mz30=PON/mz30, where PON=m/z 46:30. This makes mathematically no sense. Again all of this need to be included in the instrument section together with the quantification of PON and clearly will increase the already large uncertainty in the PON quantification during the biomass burning event.

The description of PMF and the ability of PMF to resolve the biomass burning event: In my opinion, PMF is pushed way too far in this manuscript given the quality of the data. It is well known that PMF has difficulties to resolve large individual peaks and this becomes very clear in this paper as well.

First of all this paper is clearly tailored to the AMS community, but it should at least be somewhat understandable to anybody else, especially when some of the main findings are related to the absorption of PON. None of the factors that are used such as HOA, COA or LVOOA are explained anywhere or even a reference given. What are those factors, how are they characterized, how do they relate to any of the other measured tracers and how do the ones determined in this manuscript compare to the AMS data base?

The next issue is that PMF was done in many different ways in this manuscript, many

tests were performed and a lot of these tests were subsequently discarded. So why do you describe all the tests that have not worked. As it is written, this is very confusing and really hard to follow. Furthermore, two different PMF tests were actually used in the manuscript and they are very clearly different and the results that fit the story the best were used for no apparent reason. Even the number of BBOA factors changes between the two tests.

It is very clear that PMF, even in the way as done here with separating the biomass burning event from the rest of the time series, cannot resolve the single event. In Figure 6 the combined increase of LVOOA, COA, and HOA is about twice as large as the combined BBOA_2, sPON_ME and pPON_ME signals. One might argue that during bonfire night activities such as cooking and traffic might increased as well, but certainly not to such large extents. Especially the LVOOA signal, which is larger than the BBOOA signal, has to be from biomass burning as well. So it seems that the interferences in the biomass burning event are larger than the BBOA signal itself.

Given this large uncertainty and interferences on the PMF results, it seems clearly a step too far to try to separate sPON from pPON with two different methods that have large uncertainties and are not even fully consistent with each other. The I-CIMS measures some primary and secondary ON tracers, some of which are specific to biomass burning such as nitroaromatics, why have those measurements not been used to correlate with the sPON signal? It is not clear from the text on page 9, what compounds from the I-CIMS have actually been used, except some of the inorganic tracers.

Minor Comments:

There are a large number of minor issues mostly about missing Tables, wrong numbering of Sections and typos in axis labels and such, so I only point out the two most obvious ones.

- I mentioned that before, but all the references to other sections are wrong in the manuscript. Most importantly there are references to sections that don't even exist

such as Section 3.4.1, in which supposedly the interference of m/z 30 is discussed. -
- Another glaring omission is Table 1 mentioned on page 9 line 325. This Table could
be the most important evidence to support the separation of sPON from pPON, but is
unfortunately missing.
* * *
Interactive
comment

---

## Author Comment (AC1) · 15 Nov 2017

Response to comments of Referee 1.

As a general point, we agree that PMF/ME-2 analysis was probably not a perfect representation of the aerosol, however we argue that the factorisation given represents a 'best estimate' based on the testing and validation work performed. We consider that, while BBOA concentrations could be mixed with other OA sources implying a decrease on BBOA concentrations, the represents the best estimate of the temporal trend biomass burning OA emissions. This is supported by the good correlation BBOA shows with $b_{abs\_470wb}$ r2=0.880. Thus we consider BBOA time series to be accurate enough to determine primary and secondary PON concentrations.

Through discussions outside of this review, we have considered that the term Particulate Organic Nitrate (PON) is not the most appropriate as this could include nitro compounds in addition to organic nitrates. Hence, we have decided to refer to the acronym as Particulate Organic Oxides of Nitrogen (PON) which include both nitrate and nitro organic compounds. Furthermore, because the chemistry community do not strictly consider these compounds primary, the term 'primary' here should include the qualifier in that it is not necessarily produced in the fire, but on a short enough time scale that its temporal trend is indistinguishable from the actual primary emissions, so is therefore considered 'primary' within the receptor model.

Comments from referee are in blue and response from co-authors is in black.

Major comments.

1. PMF results.

It has been challenging to perform source apportionment on a dataset including a special event with high concentrations. This study takes an important step to address this issue. However, the PMF results are still not satisfactory. My major concern is that all OA factors show significant increase during the biomass burning event (Figure 5 and 6). LV-OOA increases by ~30 μg m-3. Since fresh biomass burning unlikely contribute to LV-OOA, the increase in LV-OOA suggests PMF artifacts. The COA and HOA increase by ~8 μg m-3 and ~20 μg m-3 during the biomass burning, respectively. These enhancement magnitudes cannot be explained by the inversion at night. The enhanced concentrations of LV-OOA, COA, and HOA during the biomass burning event are likely interference from biomass burning.
Using the suggested two-step approach, a clean BBOA factor still cannot be resolved. For example, BBOA_1 is a mixed factor between LV-OOA and BBOA. Have the authors tried PMF2 solver on the whole dataset? Including the biomass burning event would be useful to get a clear BBOA factor, which helps to identify the BBOA concentration during the non-biomass burning period. However, the disadvantage of this method is that the concentrations of all OA factors would falsely decrease during the biomass burning.

We agree that during bonfire night, LV-OOA, COA and HOA may be mixed with BBOA concentrations. We mention in section 4.1 that this would be the case. Conclusions will be modified mentioning that even when using the methodology of first analysing the period before and after bonfire night and then analyse the bonfire night period, it was not possible to completely separate the OA sources. We run PMF and ME-2 for the whole dataset (test1). However, when we compared solutions from different tests, analysing first the period without bonfire emissions and then the bonfire night event, was best way to do source apportionment.

The following paragraph has been added to the end of section 4.1 OA source apportionment during bfo event:

Here we show the importance of performing OA source apportionment using different approaches in order to identify the best way to deconvolve OA sources. PMF and ME-2 source apportionment tools could not completely deconvolve OA sources during the bfo event. However, due to the high correlation between for $b_{abs\_470wb}$ and BBOA_2 ($r^2 = 0.880$) we consider that while BBOA_2 might not represent the total OA concentrations from the bonfire night event, it does represent the trend of OA emmited from the biomass burning.

The authors have done careful evaluation on PMF results. PMF results from two different tests (test 2 and test 2_ON) are presented, but the PMF results are different. This causes many confusions. For example, why does the mass spectrum of BBOA change between test 2 and test 2_ON (i.e., BBOA-2 vs BBOA)? Why are two BBOA factors are resolved in test 2, but only one BBOA factor in test 2_ON? Why is SV-OOA only resolved in test 2, but not in test 2_ON? I suggest the authors to present only the most reasonable/best solution in the manuscript to avoid confusion.

We agree two solutions in the manuscript to be confusing. The manuscript has been edited, leaving only one solution on the main manuscript (Test2_ON). The other solution has been moved to the appendix, as one of the objectives of this paper is to assess ME-2 under different tests in order to study its performance.

This paragraph has been added to section 3.3 OA source apportionment:

Test2_ON was the optimal 'best estimate' solution, a brief description is given here after being compared to the other tests (Section S7.2 in supplement). From this analysis, test2 was the best way to deconvolve OA sources, with the lowest parameters analysed: residuals, Q/Qexp values and Chi square. After modifying the fragmentation table, Test2_ON still shows a good performance with low parameters (Fig. S6-S8). Refer to section S7 in supplementary material for detailed information about source apportionment strategy and analysis performed to determine the optimal solution.

2.

(1) The most important issue is that the r2 values in the manuscript do not match those in table 1.

The r2 values on table 1 are the correct r2 values. The values mentioned on the paragraph were not updated from a previous version. r2 values have been updated.

(2) What's the rationale behind eqn. 6? Why is the partial slope used? In MLR3, there is no sPON_ME2. Then how is the light-absorption of sPON_ME2 evaluated?

We have expanded the explanation of the multilinear equation. There is sPON_ME2 in MLR3, the following two paragraphs have been added to explain equation 6.

$$b_{abs\_470wb} = A + B*x1 + C*x2 \tag{6}$$

When the parameter s"D" and x3 are used it means a trilinear regression was performed. MLR1 x1=BBOA, x2=PON. MLR2 x1=BBOA_2, x2=PON. MLR3 x1= BBOA_2, x2=sPON_ME2, x3=LVOOA in HSC and x3= pPON in bfo. A is the origin and the partial slopes B, C and D represent the contribution of x1 ,x2 and x3 to $b_{abs\_470wb}$, respectively.

As used in previous studies (Elser et al., 2016;Reyes-Villegas et al., 2016), multilinear regression analysis allows to determine the relationship between one parameter and two or more variables. Here we are analysing the partial slopes and origin to determine the correlation of babs_470wb with the other variables.

(3) In the abstract, it is stated that LV-OOA absorb light at 470nm over that of black carbon. Where is the justification for this conclusion?

Sorry to be ambiguous, the abstract has been edited as follows:

Our results suggest that sPON_ME2 does not absorb light at 470 nm while pPON_ME2 and LVOOA absorb light at 470 nm.

(4) Line 278, the authors state that after modifying the fragmentation table, the correlation between babs_470wb and BBOA is improved. I wonder if the improvement is mainly because that there are only one BBOA is resolved in test 2_ON, but two BBOA factors in test 2? In other words, is the improved correlation simply due to that modifying the fragmentation table somehow helps to separate the BBOA factor? What's the r2 between babs_470wb and the sum of BBOA and BBOA_1 in test 2?

[Figure]

We still consider OA source apportionment improves after modifying the fragmentation table. During the bonfire night r2 values between $b_{abs\_470wb}$ and BBOA is 0.839 before the modification and 0.880 after the modification, while r2 =0.813 is obtained between $b_{abs\_470wb}$ and the sum of BBOA+BBOA_1. The r2 = 0.813 will be mentioned in the manuscript

(5) Line 374, the authors need to be cautious that not all organic nitrates can absorb light. Most identified light-absorbing organic nitrate are nitro aromatic compounds (Lin et al., 2016; Mohr et al., 2013).

We have decided to change our definition of PON from "particulate organic nitrate" to "particulate organic oxides of nitrogen" which will involve both nitro and nitrate organic compounds.

3. The separation of primary PON (pPON) and secondary PON (sPON).
(1) How do the authors identify pPON and sPON? In Figure 6, the sPON_ME2 has more evident signal at m/z 30 than pPON. What does the mass spectrum of organics that associate with sPON look like? Where are the organic signals associated with pPON_ME2 from? Fresh biomass burning? More discussions regarding pPON_ME2 and sPON_ME2 are required.

The following paragraph has been added to the end of the section 3.3 OA source apportionment:

These two PON factors may have different sources; one may be secondary (sPON_ME2) as it follows the trend of LVOOA concentrations (Fig. 5.a) and the other primary (pPON_ME2) which has similar trend as BBOA (Fig. 5.b). Further details about pPON_ME2 and sPON_ME2 nature will be explored in section 4.2.

Moreover, the following paragraph has been added to the end of section 4.3:

During the bfo event, pPON_ME2 showed high r2 values with carbon monoxide (0.78) as well as hydrogen cyanide (0.77), Methylformamide (0.65) and Dimethylformamide (0.63) which are typical primary pollutants related to combustion processes [(Borduas et al., 2015) and references therein]. sPON_ME2 showed low correlations with ClNO2 (0.52) and ClNO3 (0.53). High r2 values were also observed during LC episode between ClNO2 - ClNO3 and LVOOA (0.67 - 0.66) and sPON (0.74 - 0.69) proving their secondary origin.

(2) The authors use two methods to differentiate pPON and sPON. However, there are discrepancies in the results (Figure 7 vs. 8). For example, pPON_ME2 decreases slower than BBOA in Figure 8. Could the authors directly compare the results from these two methods (i.e., scatter plot)?

This information will be added to supplement

Two methods have been used to determine primary and secondary PON. In the following plots we can see primary PON comparison has a good correlation with a pearson value of 0.7 while secondary PON comparison shows a different behaviour between them.

[Figure]

Figure S13: PON comparison for the two methods used.

(3) Previous studies have attempted to run PMF analysis on combined organic and nitrate mass spectra (Sun et al., 2012; Xu et al., 2015). The authors should compare to other literature.

This paragraph has been added to methodology in the section were we describe ME-2 PON analysis:

Previous studies have quantified PON concentrations from AMS-PMF analysis to both rural and urban measurements (Sun et al., 2012;Hao et al., 2014;Xu et al., 2015;Zhang et al., 2016).

This paragraph has been added to section 4.2 PON Primary/secondary:

Primary and secondary sources of PON have been previously identified from AMS-PMF analysis; Hao et al. (2014) identified PON to be secondary of nature, produced from the interaction between forest and urban emissions, while Zhang et al. (2016) determined PON to be related to primary combustion sources.

Minor Comments

1. Line 68. It should be "Ng et al., 2017".

The citation has been edited.

2. Line 212. What's the NO2+/NO+ value of organic nitrate used in this study? This information should be mentioned in the main text.

The following paragraphs have been edited in section 2.2.2 Particulate Organic Oxides of Nitrogen (PON):

Equation 5 calculates the PON fraction ($X_{PON}$), using the signals at m/z 30 and m/z 46 to calculate m/z ratios 46:30 from AMS measurements ($R_{meas}$), from ammonium nitrate calibrations ($R_{cal}$), and from organic nitrogen ($R_{ON}$) to quantify PON concentrations.

$$X_{PON} = \frac{(R_{meas}-R_{Cal})(1+R_{ON})}{(R_{ON}-R_{cal})(1+R_{meas})} \tag{5}$$

Where ratios from ammonium nitrate calibrations Rcal = 0.5; Rmeas = m/z 46:30 ratio from measurements; m/z 46:30 ratio from ON RON = 0.1, Following Kostenidou et al. (2015) consideration, $R_{ON}$ = 0.1 was calculated as the minimum m/z 46:30 ratio observed. RON value of 0.1 has been used in previous studies (Kiendler-Scharr et al., 2016;Tiitta et al., 2016).

$$PON = X_{PON} * NO_3^- \tag{6}$$

Finally, equation 6 calculates PON concentrations [µg.m-3] where NO3- is the total nitrate measured by the cToF-AMS.

3. Line 227. The mass spectrum of factor 4 is very similar to that of SV-OOA in step a. Then how do the authors justify "SV-OOA" in step a?

This figure has been moved to the appendix A. In this figure, we consider factor 4 mass spectra to be SVOOA in both (a) and (b).

4. Line 392, please cite Washenfelder et al. (2015), which showed that biomass burning OA is light absorbing.
The citation has been added to the manuscript.

References

Borduas, N., da Silva, G., Murphy, J. G., and Abbatt, J. P. D.: Experimental and theoretical understanding of the gas phase oxidation of atmospheric amides with oh radicals: Kinetics, products, and mechanisms, The Journal of Physical Chemistry A, 119, 4298-4308, 10.1021/jp503759f, 2015.
Elser, M., Huang, R. J., Wolf, R., Slowik, J. G., Wang, Q., Canonaco, F., Li, G., Bozzetti, C., Daellenbach, K. R., Huang, Y., Zhang, R., Li, Z., Cao, J., Baltensperger, U., El-Haddad, I., and André, P.: New insights into pm2.5 chemical composition and sources in two major cities in china during extreme haze events using aerosol mass spectrometry, Atmos Chem Phys, 16, 3207-3225, 10.5194/acp-16-3207-2016, 2016.
Hao, L. Q., Kortelainen, A., Romakkaniemi, S., Portin, H., Jaatinen, A., Leskinen, A., Komppula, M., Miettinen, P., Sueper, D., Pajunoja, A., Smith, J. N., Lehtinen, K. E. J., Worsnop, D. R., Laaksonen, A., and Virtanen, A.: Atmospheric submicron aerosol composition and particulate organic nitrate formation in a boreal forestland-urban mixed region, Atmos Chem Phys, 14, 13483-13495, 10.5194/acp-14-13483-2014, 2014.
Kiendler-Scharr, A., Mensah, A. A., Friese, E., Topping, D., Nemitz, E., Prevot, A. S. H., Äijälä, M., Allan, J., Canonaco, F., Canagaratna, M., Carbone, S., Crippa, M., Dall Osto, M., Day, D. A., De Carlo, P., Di Marco, C. F., Elbern, H., Eriksson, A., Freney, E., Hao, L., Herrmann, H., Hildebrandt, L., Hillamo, R., Jimenez, J. L., Laaksonen, A., McFiggans, G., Mohr, C., O'Dowd, C., Otjes, R., Ovadnevaite, J., Pandis, S. N., Poulain, L., Schlag, P., Sellegri, K., Swietlicki, E., Tiitta, P., Vermeulen, A., Wahner, A., Worsnop, D., and Wu, H. C.: Ubiquity of organic nitrates from nighttime chemistry in the european submicron aerosol, Geophys Res Lett, 43, 7735-7744, 10.1002/2016gl069239, 2016.
Kostenidou, E., Florou, K., Kaltsonoudis, C., Tsiflikiotou, M., Vratolis, S., Eleftheriadis, K., and Pandis, S. N.: Sources and chemical characterization of organic aerosol during the summer in the eastern mediterranean, Atmos. Chem. Phys., 15, 11355-11371, 10.5194/acp-15-11355-2015, 2015.
Reyes-Villegas, E., Green, D. C., Priestman, M., Canonaco, F., Coe, H., Prévôt, A. S. H., and Allan, J. D.: Organic aerosol source apportionment in london 2013 with me-2: Exploring the solution space with annual and seasonal analysis, Atmos. Chem. Phys., 16, 15545-15559, 10.5194/acp-16-15545-2016, 2016.
Sun, Y. L., Zhang, Q., Schwab, J. J., Yang, T., Ng, N. L., and Demerjian, K. L.: Factor analysis of combined organic and inorganic aerosol mass spectra from high resolution aerosol mass spectrometer measurements, Atmos Chem Phys, 12, 8537-8551, DOI 10.5194/acp-12-8537-2012, 2012.
Tiitta, P., Leskinen, A., Hao, L., Yli-Pirilä, P., Kortelainen, M., Grigonyte, J., Tissari, J., Lamberg, H., Hartikainen, A., Kuuspalo, K., Kortelainen, A. M., Virtanen, A., Lehtinen, K. E. J., Komppula, M., Pieber, S., Prévôt, A. S. H., Onasch, T. B., Worsnop, D. R., Czech, H., Zimmermann, R., Jokiniemi, J., and Sippula, O.: Transformation of logwood combustion emissions in a smog chamber: Formation of secondary organic aerosol and changes in the primary organic aerosol upon daytime and nighttime aging, Atmos. Chem. Phys., 16, 13251-13269, 10.5194/acp-16-13251-2016, 2016.
Xu, L., Suresh, S., Guo, H., Weber, R. J., and Ng, N. L.: Aerosol characterization over the southeastern united states using high-resolution aerosol mass spectrometry: Spatial and seasonal variation of aerosol composition and sources with a focus on organic nitrates, Atmos. Chem. Phys., 15, 7307-7336, 10.5194/acp-15-7307-2015, 2015.
Zhang, J. K., Cheng, M. T., Ji, D. S., Liu, Z. R., Hu, B., Sun, Y., and Wang, Y. S.: Characterization of submicron particles during biomass burning and coal combustion periods in beijing, china, Science of The Total Environment, 562, 812-821, 10.1016/j.scitotenv.2016.04.015, 2016.

---

## Author Comment (AC2) · 15 Nov 2017

Response to comments of Referee 2.

Comments from referee are in blue and response from co-authors is in black.

Major comments.

Quantification of PON from biomass burning using a CToF-AMS: The m/z 46:30 ratio is used for the quantification of PON, which is based on a method done by Farmer et al 2010 that used an HR-AMS and did not include any biomass burning emissions. The large fraction of OA from biomass burning certainly will produce a large CH2O+ contribution and will make the quantification of PON very different compared to the Farmer et al paper. On page 7 line 250 the authors say that the m/z 30 interference is likely small, but this might be very different in biomass burning and needs to be shown or given a reference. In addition it is written on page 6 line 206 that the interference of CH2O+ is discussed in section 3.4.1, but no such section exists in the paper. So it is not clear, how the authors deal with the m/z 30 interference and how large the uncertainty or error estimate on the quantification of PON is. In addition in the later section, where the primary versus secondary PON is discussed, it is certainly possible that the m/z 30 interference is different as the composition changes during the later part of the night, when the authors claim that they observed secondary PON. A section needs to be added, where the interference is clearly explained and the effect needs to be quantified. This should result in an uncertainty range for the m/z 30:46 ratio and error estimate for PON. All of this should be added to the instrumentation section, where there needs to be a description of the AMS added as well that includes a discussion on the calibration for NH4NO3 and, if available, for PON compounds. Here it should be made clear what m/z 30:46 ratio NH4NO3 has in this particular instrument? The resulting error estimate needs to be taken into account for the following discussions.

Section 2.2.2 in methods has been edited as follows:

 2.2.2 Particulate Organic Oxides of Nitrogen (PON)

Concentrations of PON were calculated following the method proposed by Farmer et al. (2010) and the considerations used by Kiendler-Scharr et al. (2016). This method has been previously used in studies looking at aerosols from biomass burning (Tiitta et al., 2016;Zhu et al., 2016;Florou et al., 2017). Equation 5 calculates the PON fraction ($X_{PON}$), using the signals at m/z 30 and m/z 46 to calculate m/z ratios 46:30 from AMS measurements ($R_{meas}$), from ammonium nitrate calibrations ($R_{cal}$), and from organic nitrogen ($R_{ON}$) to quantify PON concentrations.

$$X_{PON} = \frac{(R_{meas} - R_{Cal})(1 + R_{ON})}{(R_{ON} - R_{cal})(1 + R_{meas})}$$  (5)

Where ratios from ammonium nitrate calibrations $R_{cal} = 0.5$; $R_{meas}$ = m/z  46:30 ratio from measurements; m/z 46:30 ratio from ON $R_{ON} = 0.1$, Following Kostenidou et al. (2015) consideration, $R_{ON} = 0.1$ was calculated as the minimum m/z  46:30 ratio observed. $R_{ON}$ value of 0.1 has been used in previous studies (Kiendler-Scharr et al., 2016;Tiitta et al., 2016).

$$PON = X_{PON} * NO_3^-$$  (6)

Finally, equation 6 calculates PON concentrations [$\mu g.m^{-3}$] where $NO_3^-$ is the total nitrate measured by the cToF-AMS. The method proposed by Farmer et al. (2010) is based on HR-ToF-AMS measurements were m/z 30  represents $NO^+$ ion and m/z 46 $NO_2^+$ ion while the cToF-AMS gives unit mass resolution mass spectra information, hence, there is the possibility to have interference of $CH_2O^+$ ion at m/z 30. However, when analysing mass spectra from previous laboratory and ambient studies using HR-ToF-AMS to investigate biomass burning emissions, we can confirm that the signal of $CH_2O^+$ at m/z 30 is low compared to signals at m/z's 29 and 31, while in this study m/z 30 is the main signal (Fig. 5.c). Hence, in this study an interference of $CH_2O^+$ at m/z 30 is unlikely and if there were any interference of $CH_2O^+$ it would be negligible. Table S1 in

supplement shows m/z 30/29 and 30/31 from previous laboratory and ambient studies investigating biomass burning emissions.

Another possible interference would be the presence of mineral nitrates at m/z 30 (e.g. $KNO_3$ and $NaNO_3$). However, mineral nitrate salts tend to be large particles (Allan et al., 2006;Chakraborty et al., 2016) and also have low vaporisation efficiency (Drewnick et al., 2015), which makes it unlikely to be measured by the AMS in large quantities.

Table S1. $CH_2O^+$ signals at m/z 29, 30 and 31 from HR-ToF-AMS data of previous studies. Comparison of m/z ratios 30/29 and 30/31 with values found in this study.

| | Reference | 30/29 | 30/31 | m/z 29 | m/z 30 | m/z 31 | Notes |
|---|---|---|---|---|---|---|---|
| ambient | This study | 4.38 | 35.00 | 0.08 | 0.35 | 0.01 | sPON_ME2 |
| | | 1.42 | 8.50 | 0.06 | 0.09 | 0.01 | pPON_ME2 |
| | (Aiken et al., 2010) | 0.16 | 0.32 | 0.05 | 0.008 | 0.025 | pine burn |
| | | 0.20 | 0.45 | 0.045 | 0.009 | 0.02 | BBOA Mex |
| | (Collier et al., 2016) | 0.25 | 0.56 | 4 | 1 | 1.8 | Ground plume |
| | | 0.20 | 0.60 | 3 | 0.6 | 1 | Ground plume |
| | | 0.23 | 0.67 | 3.5 | 0.8 | 1.2 | aircraft plume |
| | | 0.25 | 1.25 | 4 | 1 | 0.8 | aircraft plume |
| | (Zhou et al., 2017) | 0.18 | 0.88 | 8 | 1.4 | 1.6 | no bb |
| | | 0.32 | 0.95 | 6 | 1.9 | 2 | bb inf |
| | | 0.30 | 0.90 | 6 | 1.8 | 2 | bb plm |
| Laboratory-based | (He et al., 2010) | 0.25 | 0.75 | 0.06 | 0.015 | 0.02 | Fir (diluted/cooled) |
| | | 0.21 | 0.68 | 0.07 | 0.015 | 0.022 | pine burn |
| | | 0.20 | 0.56 | 0.05 | 0.01 | 0.018 | Willow |
| | | 0.30 | 0.90 | 0.06 | 0.018 | 0.02 | Wattle |
| | | 0.30 | 0.90 | 0.06 | 0.018 | 0.02 | SugaCaneLeave |
| | | 0.30 | 0.08 | 0.05 | 0.015 | 0.2 | Rice Straw |
| | (Heringa et al., 2011) | 0.25 | 0.67 | 4 | 1 | 1.5 | poa |
| | | 0.25 | 0.50 | 4 | 1 | 2 | 5h aging |
| | (Ortega et al., 2013) | 0.15 | 0.50 | 13 | 2 | 4 | start (oak) |
| | | 0.20 | 0.50 | 50 | 10 | 20 | aged (oak) |
| | | 0.04 | 0.05 | 250 | 10 | 220 | start (pine) |
| | | 0.07 | 0.10 | 270 | 20 | 200 | aged (pine) |
| | (Corbin et al., 2015b) | 0.20 | 0.80 | 4 | 0.8 | 1 | start |
| | | | 0.83 | | 0.05 | 0.06 | flaming |
| | (Corbin et al., 2015a) | | 0.50 | | 0.01 | 0.02 | Filtered and Oxid |
| | | | 0.50 | | 0.01 | 0.02 | Oxidized |
| | | 0.25 | 0.50 | 0.04 | 0.01 | 0.02 | Primary |
| | (Bruns et al., 2015) | 0.43 | 6.00 | 0.07 | 0.03 | 0.005 | OH and UV exp. |
| | | 0.34 | 1.00 | 0.065 | 0.022 | 0.022 | OH and UV exp. |
| | | 0.40 | 1.00 | 0.045 | 0.018 | 0.018 | OH and UV exp. |
| | | 0.34 | 1.00 | 0.065 | 0.022 | 0.022 | OH and UV exp. |
| | | 0.40 | 1.00 | 0.045 | 0.018 | 0.018 | OH and UV exp. |
| | | 0.23 | 1.00 | 0.048 | 0.011 | 0.011 | OH and UV exp. |
| | | 0.20 | 1.00 | 0.04 | 0.008 | 0.008 | OH and UV exp. |
| | | 0.25 | 1.00 | 0.048 | 0.012 | 0.012 | OH and UV exp. |

These paragraphs have been added to describe Table S1 in the supplement.

Table S1 shows $CH_2O^+$ signals at m/z's 29, 30, and 31 from HR-ToF-AMS studies. It is possible to observe the low $CH_2O^+$ contribution to m/z 30 with 30/29 ratios between 0.01-0.40. The high values of $0.4 - 6$ were observed when exposing aerosols to OH and UV. We can also see that 30/31 30/29 ratios do not show variations during and after biomass burning events or during fresh and aged emissions (Ortega et al., 2013;Corbin et al., 2015a;Corbin et al., 2015b), suggesting there is not substantial $CH_2O^+$ variability over the biomass burning process.

In this study, a large contribution of m/z 30 signal to the mass spectra was observed with both sPON and pPON with 30/29 ratios (4.38 and 1.42 respectively) and 30/31 ratios (35.0 and 8.5 respectively) higher than unity. Showing that a $CH_2O^+$ interference at m/z30 would be unlikely.

For biomass burning the fragmentation tables need to be modified, which the authors do later in the paper. It does not make sense to me to run PMF on not-corrected data as was done in Section 3.3, when you know you are using incorrect data.

We agree on showing both solutions in the main manuscript is confusing. The PMF solution without modifying the fragmentation table has been moved to the appendix. Part of this paper is to explore different ways to run ME-2 and evaluate its performance under different conditions, hence we consider important to show this solution on the appendix.

The section on the FragPanel modification is very specific to AMS users and all the three bullet points cannot be understood by anybody else without explaining all the abbreviations and acronyms. Especially sentences like on page 7 line 238: time series of PON:mz30 were calculated with the equation PON:mz30=PON/mz30, where PON=m/z 46:30. This makes mathematically no sense. Again all of this need to be included in the instrument section together with the quantification of PON and clearly will increase the already large uncertainty in the PON quantification during the biomass burning event.

The following paragraphs have been added to the end of section 2.2.3 Multilinear engine 2 (ME-2):

PON may exhibit covariance with other types of OA, thus their inclusion in the source apportionment analysis may give a more complete factorisation and highlight their co-emission with other OA types. Therefore, a different experiment was designed by modifying the fragmentation table, through the AMS analysis toolkit 1.56, in order to identify a PON source. The fragmentation table contains the different chemical species measured by the AMS, with each row representing m/z for specific species and where the user can define peaks that exist in each species' partial mass spectrum and their dependency on other peaks (Allan et al., 2004). The following steps were performed to modify the fragmentation table:

• Time series of a new ratio named RON_30 is calculated by RON_30 = PON/mz30, were PON is the time series calculated in section 2.2.2 and mz30 is the time series of the signal at m/z=30 measured by the AMS.

• Using the AMS analysis toolkit; the fragmentation table is modified, in the column frag_Organic at the m/z 30, by multiplying RON_30*30. See figure S4 in supplement for a screenshot of the fragmentation table.

• PMF inputs are generated to be used in SoFi software.

Figure S4 has been added to the supplement to show how fragmentation table has been modified.

[Figure]

Figure S4: Modifying fragmentation table to add PON to PMF analysis.

The description of PMF and the ability of PMF to resolve the biomass burning event:

In my opinion, PMF is pushed way too far in this manuscript given the quality of the data. It is well known that PMF has difficulties to resolve large individual peaks and this becomes very clear in this paper as well.

As mentioned to Referee 1, we agree that, during bonfire night, LV-OOA, COA and HOA may be mixed with BBOA concentrations. We mentioned in section 4.1 that this would be the case. Conclusions will be modified mentioning that even when using the methodology of first analysing the period before and after bonfire night and then analyse the bonfire night period, it was not possible to completely separate the OA sources. However, we consider that, while BBOA concentrations would be mixed with other OA sources implying a decrease on BBOA concentrations, the actual BBOA trend remains and BBOA factor is considered to be representative of biomass burning OA emissions. This is supported by the good correlation BBOA shows with $b_{abs\_470wb}$ $r^2$=0.880, a marker of emissions from biomass burning. Thus we consider BBOA time series to be accurate enough to determine primary and secondary PON concentrations.

The following paragraph has been added to the end of section 4.1 OA source apportionment:

Here is shown the importance of performing OA source apportionment using different approaches in order to identify the best way to deconvolve OA sources. PMF and ME-2 source apportionment tools could not completely deconvolve OA sources during the bfo event. However, due to the high correlation between for $b_{abs\_470wb}$ and BBOA_2 ($r^2$ = 0. 880) we consider that while BBOA_2 might not represent the total OA concentrations from the bonfire night event, it does represent the trend of OA emmited from the biomass burning.

First of all this paper is clearly tailored to the AMS community, but it should at least be somewhat understandable to anybody else, especially when some of the main findings are related to the absorption of PON. None of the factors that are used such as HOA, COA or LVOOA are explained anywhere or even a reference given. What are those factors, how are they characterized, how do they relate to any of the other measured tracers and how do the ones determined in this manuscript compare to the AMS data base?

We have added a more understandable description of the OA sources with their respective references, The following paragraph has been edited in the section 3.3 OA source apportionment:

Two steps were involved in Test2_ON: in step a, PMF/ME-2 were run for the event before and after the bonfire night (named as not bonfire event, nbf). In Step b, mass spectra from the solution identified in step a were used as TP, to analyse the bonfire-only (bfo) event. Finally, both solutions (nbf and bfo) were merged for further analysis. Different OA sources were identified in Test2_ON (Fig. 5), five sources were identified during nbf event: biomass burning OA (BBOA), hydrocarbon-like OA (HOA), cooking OA (COA), secondary particulate organic oxides of nitrogen (sPON_ME2) and low volatility OA (LVOOA). These sources are identified by characteristic peaks at their mass spectrum; BBOA, which is generated during the combustion of biomass, has a peak at m/z 60, related to levoglucosan (Alfarra et al., 2007); HOA, related to traffic emissions, presents high signals at m/z 55 and m/z 57 typical of aliphatic hydrocarbons (Canagaratna et al., 2004); COA, emitted from food cooking activities, is similar to HOA with a higher m/z 55 and lower m/z 57 (Allan et al., 2010;Slowik et al., 2010;Mohr et al., 2012); LVOOA, identified as a secondary organic aerosol, has a high signal at m/z 44 dominated by the $CO_2^+$ ion (Ng et al., 2010); sPON_ME2 has a strong signal at m/z 30 and it has been identified to be secondary as it follows the same trend as LVOOA (Figure 5.a). In the case of the bfo event six different sources were identified: BBOA, HOA, COA, LVOOA and two factors with peaks at m/z30, which is related to PON (Sun et al., 2012). These two PON factors may have different sources; one may be secondary (sPON_ME2) and the other primary (pPON_ME2) which has similar trend as BBOA (Fig. 5.b). Further details about pPON_ME2 and sPON_ME2 nature will be explored in section 4.2.

The next issue is that PMF was done in many different ways in this manuscript, many tests were performed and a lot of these tests were subsequently discarded. So why do you describe all the tests that have not worked. As it is written, this is very confusing and really hard to follow. Furthermore, two different PMF tests were actually used in the manuscript and they are very clearly different and the results that fit the story the best were used for no apparent reason. Even the number of BBOA factors changes between the two tests.

One of the objectives of this paper is to explore different ways to perform source apportionment, hence we consider important to mention the different tests perform. However, we agreed that mentioning both solutions in the main text is confusing. Hence, we have modified the manuscript by only explaining in the main text the optimal solution (Test2_ON) and moving the other solution to the appendix.

It is very clear that PMF, even in the way as done here with separating the biomass burning event from the rest of the time series, cannot resolve the single event. In Figure 6 the combined increase of LVOOA, COA, and HOA is about twice as large as the combined BBOA_2, sPON_ME and pPON_ME signals. One might argue that during bonfire night activities such as cooking and traffic might increased as well, but certainly not to such large extents. Especially the LVOOA signal, which is larger than the BBOOA signal, has to be from biomass burning as well. So it seems that the interferences in the biomass burning event are larger than the BBOA signal itself.
Given this large uncertainty and interferences on the PMF results, it seems clearly a step too far to try to separate sPON from pPON with two different methods that have large uncertainties and are not even fully consistent with each other. The I-CIMS measures some primary and secondary ON tracers, some of which are specific to biomass burning such as nitroaromatics, why have those measurements not been used to correlate with the sPON signal? It is not clear from the text on page 9, what compounds from the I-CIMS have actually been used, except some of the inorganic tracers.

This is the correct table, which is in the supplement:
Table S4: $R^2$ values between OA factors and CIMS measurements.

| Formula | Name | BBOA | | | | | COA | | | | sPON | | | | LVOOA | | | | pPON |
|---|---|---|---|---|---|---|---|---|---|---|---|---|---|---|---|---|---|---|---|
| | | ALL | HSC | LC | bfo | WL | ALL | LC | bfo | WL | ALL | LC | bfo | WL | ALL | LC | bfo | WL | bfo |
| C4H6O2 | methacrylic acid | 0.89 | | | 0.92 | 0.53 | 0.64 | | 0.77 | 0.48 | | | | | 0.78 | | 0.82 | | 0.52 |
| C3H4O2 | Acid_Acrylic | 0.85 | | | 0.90 | 0.65 | 0.62 | | 0.70 | 0.43 | | | | 0.48 | 0.79 | | 0.88 | | |
| H2COH2O | methylhydroperoxide | 0.78 | | | 0.90 | | 0.54 | | 0.69 | | | | | | 0.66 | | 0.85 | | |
| C6H6O | Phenol | 0.89 | | | 0.89 | | 0.59 | | 0.73 | | | | | | 0.75 | | 0.73 | | 0.57 |
| C7H6O2 | Benzoic acid | 0.89 | | 0.57 | 0.89 | 0.86 | 0.65 | | 0.83 | 0.45 | | 0.71 | | 0.73 | 0.67 | 0.72 | 0.64 | 0.58 | 0.57 |
| C2H5NO | Methylformamide | 0.88 | | | 0.89 | 0.47 | 0.61 | | 0.79 | | | | | | 0.65 | | 0.67 | 0.56 | 0.65 |
| C2H3NO | Methyl isocyanate | 0.89 | 0.49 | 0.44 | 0.89 | 0.71 | 0.55 | | 0.66 | | | | | 0.50 | 0.85 | | 0.88 | | |
| C5H10O2 | Pentanoic acid | 0.77 | | | 0.87 | | 0.60 | | 0.76 | | | | | | 0.54 | | 0.66 | | |
| HNO2 | nitrous acid | 0.81 | | | 0.86 | 0.66 | 0.59 | | 0.84 | | | | | 0.57 | 0.61 | | 0.66 | | 0.70 |
| CH2O2 | formic acid | 0.52 | | | 0.86 | | | | 0.62 | | | | | | 0.58 | | 0.88 | | |
| C3H7NO | Dimethylformamide | 0.80 | | | 0.85 | | 0.59 | | 0.76 | | | | | | 0.56 | | 0.63 | 0.60 | 0.63 |
| C3H6O2 | propionic acid | 0.87 | | 0.67 | 0.85 | 0.72 | 0.53 | 0.45 | 0.62 | | | 0.41 | | 0.67 | 0.78 | | 0.78 | 0.63 | |
| C2H5N3O2 | C2H5N3O2 | | | | 0.83 | | | | 0.77 | | | | | | | | 0.59 | | |
| CHNO | Isocyanic acid | 0.86 | | 0.64 | 0.83 | | 0.56 | | 0.68 | | | 0.81 | | | 0.84 | 0.80 | 0.86 | | 0.47 |
| C4H6O4 | succinic acid | | | | 0.83 | | | | 0.71 | | | | | | | | 0.60 | | |
| C6H6O3 | trihydroxybenzene | 0.83 | 0.48 | 0.72 | 0.82 | 0.85 | 0.62 | | 0.79 | 0.42 | | 0.75 | | 0.71 | 0.59 | 0.82 | 0.54 | 0.59 | 0.49 |
| C4H8O2 | butyric acid | | | | 0.80 | | | | 0.58 | | | | | | | | 0.76 | | |
| C2H2NO3 | C2H2NO3 | 0.61 | | | 0.79 | | 0.48 | | 0.56 | | | 0.49 | | | 0.63 | | 0.90 | | |
| HO2H2O | HO2H2O | 0.53 | | | 0.77 | | | | 0.63 | | | | | | | | 0.70 | | |
| CHN | Hydrogen cyanide | 0.80 | | 0.66 | 0.76 | 0.84 | 0.57 | 0.36 | 0.70 | | | 0.60 | | 0.74 | 0.62 | 0.69 | 0.61 | 0.54 | 0.77 |
| C6H6O2 | Catechol | 0.73 | | | 0.73 | | 0.44 | | 0.56 | | | | | | 0.63 | | 0.62 | | |
| C7H8O | Cresol | 0.79 | | | 0.72 | | 0.50 | | 0.59 | | | | | | 0.59 | | 0.51 | | 0.65 |
| C3H4O4 | Malonic acid | | | | 0.69 | | | | 0.50 | | | | | | | | 0.52 | 0.54 | |
| C7H8O2 | guaiacol | 0.63 | | | 0.62 | 0.78 | | | 0.45 | 0.43 | | | | 0.62 | 0.58 | | 0.57 | | |
| C2H4O3 | Glycolic Acid | | | | 0.62 | | 0.42 | | 0.63 | | | | | | | | | | |
| CNO | anion isocyanate | 0.66 | | 0.61 | 0.61 | | 0.48 | | 0.50 | | | 0.81 | | | 0.74 | 0.76 | 0.74 | | |
| C3H7NO2 | L-Alanine | | | | 0.54 | | | | 0.64 | | | | | | | | | | 0.65 |
| *NO | | 0.40 | | | 0.63 | | | | 0.59 | | | | | | | | 0.58 | | 0.46 |
| *NO2 | | | | | 0.45 | 0.51 | | | 0.41 | | | | 0.50 | | | | 0.54 | | |
| *Nox | | | | | 0.60 | 0.47 | | | 0.57 | | | | | | | | 0.59 | | |
| *CO | | 0.79 | 0.55 | | 0.81 | 0.67 | 0.64 | | 0.80 | 0.42 | | | | 0.48 | 0.58 | | 0.56 | | 0.78 |
| *SO2 | | | | | 0.63 | | | | 0.57 | | | | | | | | 0.52 | | 0.72 |
| ClNO3 | Chlorine nitrate | | | 0.45 | | | | | | | 0.45 | 0.69 | 0.53 | | | | 0.66 | | |
| ClNO2 | nitryl chloride | | | 0.47 | | | | | | | | 0.74 | 0.52 | | | | 0.67 | | |
| Cl2 | Chlorine | | | | | | | | | | | | 0.51 | | | | | | 0.44 |
| C6H5NO3 | nitrophenol | | 0.41 | | | | | | | | | | | | | | | 0.55 | |

ALL = all dataset, LC = low concentrations, bfo = bonfire night, WL = winter-like.

The following paragraph has been added at the end of the section 4.3 OA factors and CIMS correlations, which supports the primary and secondary nature of PON:

during the bfo event, pPON_ME2 showed high $r^2$ values with carbon monoxide (0.78) as well as hydrogen cyanide (0.77), Methylformamide (0.65) and Dimethylformamide (0.63) which are typical primary pollutants related to combustion processes [(Borduas et al., 2015) and references therein]. sPON_ME2 showed low correlations with $ClNO_2$ (0.52) and $ClNO_3$ (0.53). High $r^2$ values were also observed during LC episode between $ClNO_2$ - $ClNO_3$ and LVOOA (0.67 - 0.66) and sPON (0.74 - 0.69) proving their secondary origin. $Cl_2$, which has been previously identified to be related to both primary and secondary sources (Faxon et al., 2015), shows low correlations with pPON_ME2 (0.44) during bfo event and sPON_ME2 (0.55) during LC event.

Minor Comments:

There are a large number of minor issues mostly about missing Tables, wrong numbering of Sections and typos in axis labels and such, so I only point out the two most obvious ones.

We are sorry for not updating the numbers and labelling in the last version of the manuscript before submitting it to ACPD. A full update of numbering to tables and figures has been performed.

- I mentioned that before, but all the references to other sections are wrong in the manuscript. Most importantly there are references to sections that don't even exist such as Section 3.4.1, in which supposedly the interference of m/z 30 is discussed. –

The interference of m/z 30 is being discussed in section 2.2.2

- Another glaring omission is Table 1 mentioned on page 9 line 325. This Table could be the most important evidence to support the separation of sPON from pPON, but is unfortunately missing.

This table is in supplement as Table S4.

References

Aiken, A. C., De Foy, B., Wiedinmyer, C., Decarlo, P. F., Ulbrich, I. M., Wehrli, M. N., Szidat, S., Prevot, A. S. H., Noda, J., Wacker, L., Volkamer, R., Fortner, E., Wang, J., Laskin, A., Shutthanandan, V., Zheng, J., Zhang, R., Paredes-Miranda, G., Arnott, W. P., Molina, L. T., Sosa, G., Querol, X., and Jimenez, J. L.: Mexico city aerosol analysis during milagro using high resolution aerosol mass spectrometry at the urban supersite (t0)-part 2: Analysis of the biomass burning contribution and the non-fossil carbon fraction, Atmos Chem Phys, 10, 5315-5341, 10.5194/acp-10-5315-2010, 2010.
Alfarra, M. R., Prevot, A. S. H., Szidat, S., Sandradewi, J., Weimer, S., Lanz, V. A., Schreiber, D., Mohr, M., and Baltensperger, U.: Identification of the mass spectral signature of organic aerosols from wood burning emissions, Environmental Science & Technology, 41, 5770-5777, Doi 10.1021/Es062289b, 2007.
Allan, J. D., Delia, A. E., Coe, H., Bower, K. N., Alfarra, M. R., Jimenez, J. L., Middlebrook, A. M., Drewnick, F., Onasch, T. B., Canagaratna, M. R., Jayne, J. T., and Worsnop, D. R.: A generalised method for the extraction of chemically resolved mass spectra from aerodyne aerosol mass spectrometer data, J Aerosol Sci, 35, 909-922, DOI 10.1016/j.jaerosci.2004.02.007, 2004.
Allan, J. D., Alfarra, M. R., Bower, K. N., Coe, H., Jayne, J. T., Worsnop, D. R., Aalto, P. P., Kulmala, M., Hyötyläinen, T., Cavalli, F., and Laaksonen, A.: Size and composition measurements of background aerosol and new particle growth in a finnish forest during quest 2 using an aerodyne aerosol mass spectrometer, Atmos. Chem. Phys., 6, 315-327, 10.5194/acp-6-315-2006, 2006.
Allan, J. D., Williams, P. I., Morgan, W. T., Martin, C. L., Flynn, M. J., Lee, J., Nemitz, E., Phillips, G. J., Gallagher, M. W., and Coe, H.: Contributions from transport, solid fuel burning and cooking to primary organic aerosols in two uk cities, Atmos Chem Phys, 10, 647-668, 2010.
Borduas, N., da Silva, G., Murphy, J. G., and Abbatt, J. P. D.: Experimental and theoretical understanding of the gas phase oxidation of atmospheric amides with oh radicals: Kinetics, products, and mechanisms, The Journal of Physical Chemistry A, 119, 4298-4308, 10.1021/jp503759f, 2015.
Bruns, E. A., Krapf, M., Orasche, J., Huang, Y., Zimmermann, R., Drinovec, L., Močnik, G., El-Haddad, I., Slowik, J. G., Dommen, J., Baltensperger, U., and Prévôt, A. S. H.: Characterization of primary and secondary wood combustion products generated under different burner loads, Atmos. Chem. Phys., 15, 2825-2841, 10.5194/acp-15-2825-2015, 2015.
Canagaratna, M. R., Jayne, J. T., Ghertner, D. A., Herndon, S., Shi, Q., Jimenez, J. L., Silva, P. J., Williams, P., Lanni, T., Drewnick, F., Demerjian, K. L., Kolb, C. E., and Worsnop, D. R.: Chase studies of particulate emissions from in-use new york city vehicles, Aerosol Science and Technology, 38, 555-573, 10.1080/02786820490465504, 2004.
Chakraborty, A., Gupta, T., and Tripathi, S. N.: Chemical composition and characteristics of ambient aerosols and rainwater residues during indian summer monsoon: Insight from aerosol mass spectrometry, Atmos Environ, 136, 144-155, 10.1016/j.atmosenv.2016.04.024, 2016.

Collier, S., Zhou, S., Onasch, T. B., Jaffe, D. A., Kleinman, L., Sedlacek, A. J., Briggs, N. L., Hee, J., Fortner, E., Shilling, J. E., Worsnop, D., Yokelson, R. J., Parworth, C., Ge, X., Xu, J., Butterfield, Z., Chand, D., Dubey, M. K., Pekour, M. S., Springston, S., and Zhang, Q.: Regional influence of aerosol emissions from wildfires driven by combustion efficiency: Insights from the bbop campaign, Environmental Science & Technology, 50, 8613-8622, 10.1021/acs.est.6b01617, 2016.

Corbin, J. C., Keller, A., Lohmann, U., Burtscher, H., Sierau, B., and Mensah, A. A.: Organic emissions from a wood stove and a pellet stove before and after simulated atmospheric aging, Aerosol Science and Technology, 49, 1037-1050, 10.1080/02786826.2015.1079586, 2015a.

Corbin, J. C., Lohmann, U., Sierau, B., Keller, A., Burtscher, H., and Mensah, A. A.: Black carbon surface oxidation and organic composition of beech-wood soot aerosols, Atmos. Chem. Phys., 15, 11885-11907, 10.5194/acp-15-11885-2015, 2015b.

Drewnick, F., Diesch, J. M., Faber, P., and Borrmann, S.: Aerosol mass spectrometry: Particle–vaporizer interactions and their consequences for the measurements, Atmos. Meas. Tech., 8, 3811-3830, 10.5194/amt-8-3811-2015, 2015.

Farmer, D. K., Matsunaga, A., Docherty, K. S., Surratt, J. D., Seinfeld, J. H., Ziemann, P. J., and Jimenez, J. L.: Response of an aerosol mass spectrometer to organonitrates and organosulfates and implications for atmospheric chemistry, Proceedings of the National Academy of Sciences of the United States of America, 107, 6670-6675, 10.1073/pnas.0912340107, 2010.

Faxon, C., Bean, J., and Ruiz, L.: Inland concentrations of cl2 and clno2 in southeast texas suggest chlorine chemistry significantly contributes to atmospheric reactivity, Atmosphere, 6, 1487, 2015.

Florou, K., Papanastasiou, D. K., Pikridas, M., Kaltsonoudis, C., Louvaris, E., Gkatzelis, G. I., Patoulias, D., Mihalopoulos, N., and Pandis, S. N.: The contribution of wood burning and other pollution sources to wintertime organic aerosol levels in two greek cities, Atmos. Chem. Phys., 17, 3145-3163, 10.5194/acp-17-3145-2017, 2017.

He, L. Y., Lin, Y., Huang, X. F., Guo, S., Xue, L., Su, Q., Hu, M., Luan, S. J., and Zhang, Y. H.: Characterization of high-resolution aerosol mass spectra of primary organic aerosol emissions from chinese cooking and biomass burning, Atmos. Chem. Phys., 10, 11535-11543, 10.5194/acp-10-11535-2010, 2010.

Heringa, M. F., DeCarlo, P. F., Chirico, R., Tritscher, T., Dommen, J., Weingartner, E., Richter, R., Wehrle, G., Prévôt, A. S. H., and Baltensperger, U.: Investigations of primary and secondary particulate matter of different wood combustion appliances with a high-resolution time-of-flight aerosol mass spectrometer, Atmos. Chem. Phys., 11, 5945-5957, 10.5194/acp-11-5945-2011, 2011.

Kiendler-Scharr, A., Mensah, A. A., Friese, E., Topping, D., Nemitz, E., Prevot, A. S. H., Äijälä, M., Allan, J., Canonaco, F., Canagaratna, M., Carbone, S., Crippa, M., Dall Osto, M., Day, D. A., De Carlo, P., Di Marco, C. F., Elbern, H., Eriksson, A., Freney, E., Hao, L., Herrmann, H., Hildebrandt, L., Hillamo, R., Jimenez, J. L., Laaksonen, A., McFiggans, G., Mohr, C., O'Dowd, C., Otjes, R., Ovadnevaite, J., Pandis, S. N., Poulain, L., Schlag, P., Sellegri, K., Swietlicki, E., Tiitta, P., Vermeulen, A., Wahner, A., Worsnop, D., and Wu, H. C.: Ubiquity of organic nitrates from nighttime chemistry in the european submicron aerosol, Geophys Res Lett, 43, 7735-7744, 10.1002/2016gl069239, 2016.

Kostenidou, E., Florou, K., Kaltsonoudis, C., Tsiflikiotou, M., Vratolis, S., Eleftheriadis, K., and Pandis, S. N.: Sources and chemical characterization of organic aerosol during the summer in the eastern mediterranean, Atmos. Chem. Phys., 15, 11355-11371, 10.5194/acp-15-11355-2015, 2015.

Mohr, C., DeCarlo, P. F., Heringa, M. F., Chirico, R., Slowik, J. G., Richter, R., Reche, C., Alastuey, A., Querol, X., Seco, R., Penuelas, J., Jimenez, J. L., Crippa, M., Zimmermann, R., Baltensperger, U., and Prevot, A. S. H.: Identification and quantification of organic aerosol from cooking and other sources in barcelona using aerosol mass spectrometer data, Atmos Chem Phys, 12, 1649-1665, DOI 10.5194/acp-12-1649-2012, 2012.

Ng, N. L., Canagaratna, M. R., Zhang, Q., Jimenez, J. L., Tian, J., Ulbrich, I. M., Kroll, J. H., Docherty, K. S., Chhabra, P. S., Bahreini, R., Murphy, S. M., Seinfeld, J. H., Hildebrandt, L., Donahue, N. M., DeCarlo, P. F., Lanz, V. A., Prevot, A. S. H., Dinar, E., Rudich, Y., and Worsnop, D. R.: Organic aerosol

components observed in northern hemispheric datasets from aerosol mass spectrometry, Atmos Chem Phys, 10, 4625-4641, DOI 10.5194/acp-10-4625-2010, 2010.

Ortega, A. M., Day, D. A., Cubison, M. J., Brune, W. H., Bon, D., de Gouw, J. A., and Jimenez, J. L.: Secondary organic aerosol formation and primary organic aerosol oxidation from biomass-burning smoke in a flow reactor during flame-3, Atmos. Chem. Phys., 13, 11551-11571, 10.5194/acp-13-11551-2013, 2013.

Slowik, J. G., Vlasenko, A., McGuire, M., Evans, G. J., and Abbatt, J. P.: Simultaneous factor analysis of organic particle and gas mass spectra: Ams and ptr-ms measurements at an urban site, Atmos Chem Phys, 10, 1969-1988, 2010.

Sun, Y. L., Zhang, Q., Schwab, J. J., Yang, T., Ng, N. L., and Demerjian, K. L.: Factor analysis of combined organic and inorganic aerosol mass spectra from high resolution aerosol mass spectrometer measurements, Atmos Chem Phys, 12, 8537-8551, 10.5194/acp-12-8537-2012, 2012.

Tiitta, P., Leskinen, A., Hao, L., Yli-Pirilä, P., Kortelainen, M., Grigonyte, J., Tissari, J., Lamberg, H., Hartikainen, A., Kuuspalo, K., Kortelainen, A. M., Virtanen, A., Lehtinen, K. E. J., Komppula, M., Pieber, S., Prévôt, A. S. H., Onasch, T. B., Worsnop, D. R., Czech, H., Zimmermann, R., Jokiniemi, J., and Sippula, O.: Transformation of logwood combustion emissions in a smog chamber: Formation of secondary organic aerosol and changes in the primary organic aerosol upon daytime and nighttime aging, Atmos. Chem. Phys., 16, 13251-13269, 10.5194/acp-16-13251-2016, 2016.

Zhou, S., Collier, S., Jaffe, D. A., Briggs, N. L., Hee, J., Sedlacek Iii, A. J., Kleinman, L., Onasch, T. B., and Zhang, Q.: Regional influence of wildfires on aerosol chemistry in the western us and insights into atmospheric aging of biomass burning organic aerosol, Atmos. Chem. Phys., 17, 2477-2493, 10.5194/acp-17-2477-2017, 2017.

Zhu, Q., He, L. Y., Huang, X. F., Cao, L. M., Gong, Z. H., Wang, C., Zhuang, X., and Hu, M.: Atmospheric aerosol compositions and sources at two national background sites in northern and southern china, Atmos. Chem. Phys., 16, 10283-10297, 10.5194/acp-16-10283-2016, 2016.

---

## Author Response (AR2)

Author's response (in blue) to co-editor comments (in black).

Comments to the Author:

The manuscript as a whole needs careful proof reading and language editing, e.g. paragraph starting line 323 is unclear due to language/grammar, repeated use of were instead of where in the manuscript, equation numbering etc.

The document has been revised and edited.

The authors should provide a general quantification of "high" versus "low" correlation for the whole manuscript and quantify the significance of an increased r2 (0.894 vs 0.861) after modifying the fragmentation table.

All the manuscript has been revised and the following considerations have been used when describing the correlations and added to the manuscript in line 319:

$r^2$ values are calculated and analysed using the following considerations: Strong correlation $r^2 \geq 0.75$, moderate correlation $0.5 < r2 < 0.75$ and low correlation $r^2 \leq 0.5$.

About the comparison of $r^2$ 0.894 vs 0.861, the following paragraph has been added to section 4.1 (line 325):

While both have strong correlations from a quantitative point of view, qualitatively, there is an improvement in BBOA_2.

The equation provided for multi-linear regression (eq 6 --> should be eq 7?) is given for the bilinear case, with the first parameter explained right after refering to trilinear regression. This is adding unnecessary confusion. Make a clear description of what is done in the multilinear regression, also using full sentences in the description on MLR 1-3 and make sure Table 1 is formated to provide an easy overview of the results.

The equation number has been edited. The description of multilinear regression has been edited for a better understanding. Table 1 has been formatted.

Right hand axis label in Figure 8 should presumably read N2O5 instead of NO2O5.

Figure 8 has been edited.

I uploaded a new version of the supplement with only one correction in section S9.

[revised manuscript text omitted]